**Soil-atmosphere exchange flux of total gaseous mercury (TGM) at subtropical and temperate forest catchments**

Jun Zhou [a, b, c, f], Zhangwei Wang [a, c, *], Xiaoshan Zhang [a, c], Charles T. Driscoll [d], Che-Jen Lin [e]

a. State Key Laboratory of Urban and Regional Ecology, Research Center for Eco-Environmental Sciences, Chinese Academy of Sciences, Beijing 100085, China.

b. Key Laboratory of Soil Environment and Pollution Remediation, Institute of Soil Science, Chinese Academy of Sciences, Nanjing 210008, China.

c. University of Chinese Academy of Sciences, Beijing 100049, China.

d. Department of Civil and Environmental Engineering, Syracuse University, 151 Link Hall, Syracuse, New York 13244, United States.

e. Center for Advances in Water and Air Quality, Lamar University, Beaumont, Texas 77710, United States.

f. Department of Environmental, Earth and Atmospheric Sciences, University of Massachusetts, Lowell, 01854, USA

* Corresponding author: Zhangwei Wang

E-mail address: wangzhw@rcees.ac.cn (Z. Wang); Phone: +86 10 62849168.

No.18 Shuangqing Road, Beijing 100085, China

First author e-mail: zhoujun@issas.ac.cn (J. Zhou); Phone: +86 25 86881319.

No.73 East Beijing Road, Nanjing 210008, China.

**Abstract:** Evasion from soil is the largest source of mercury (Hg) to the atmosphere from terrestrial ecosystems. To improve understanding of controls and in estimates of forest soil-atmosphere fluxes of total gaseous Hg (TGM), measurements were made using dynamic flux chambers (DFC) over 130 and 96 days for each of five plots at a subtropical forest and a temperate forest, respectively. At the subtropical forest the highest net soil Hg emissions were observed for an open field ($24 \pm 33$ ng $m^{-2}$ $hr^{-1}$), followed by two coniferous forest plots ($2.8 \pm 3.9$ and $3.5 \pm 4.2$ ng $m^{-2}$ $hr^{-1}$), broad-leaved forest plot ($0.18 \pm 4.3$ ng $m^{-2}$ $hr^{-1}$), and the remaining wetland site showing net deposition ($-0.80 \pm 5.1$ ng $m^{-2}$ $hr^{-1}$). At the temperate forest, the highest fluxes and net soil Hg emissions were observed for a wetland ($3.81 \pm 0.52$ ng $m^{-2}$ $hr^{-1}$) and an open field ($1.82 \pm 0.79$ ng $m^{-2}$ $hr^{-1}$), with lesser emission rates in deciduous broad-leaved forest ($0.68 \pm 1.01$ ng $m^{-2}$ $hr^{-1}$) and deciduous needle-leaved forest ($0.32 \pm 0.96$ ng $m^{-2}$ $hr^{-1}$) plots, and net deposition at an evergreen pine forest ($-0.04 \pm 0.81$ ng $m^{-2}$ $h^{-1}$). High solar radiation and temperature during summer resulted in the high Hg emissions in the subtropical forest, and the open field and evergreen pine forest at the temperate forest. At the temperate deciduous plots, the highest Hg emission occurred in spring during leaf-off period due to direct solar radiation exposure to soils. Fluxes showed strong positive relationships with solar radiation and soil temperature, and negative correlations with ambient-air TGM concentration in both subtropical and temperate forests, with area-weighted compensation points of 6.82 and 3.42 ng $m^{-3}$, respectively. The values of the compensation points suggest that the atmospheric TGM concentration can play a critical role in limiting TGM emissions from the forest floor. Climate change and land-use disturbance may increase the compensation points in both temperate and subtropical forests. Future research should focus on the role of legacy soil Hg in reemissions to the atmosphere as decreases in primary emissions drive decreases in TGM concentrations and disturbance of climate change and land use.

**Keywords:** soil-air flux of total gaseous mercury; dynamic flux chamber; compensation point; climate change; land use

## 1. Introduction

Mercury (Hg) is a persistent, bio-accumulative, toxic and well-established global contaminant (Obrist et al., 2018). Unlike other trace metals in the atmosphere, the Hg mainly exists as gaseous elemental Hg (Hg(0)), which accounts more than 90% of total gaseous Hg (TGM). Hg(0) is relatively inert and has a long atmospheric lifetime of 0.5–1 year, which allows for long range transport (Kamp et al., 2018;Slemr et al., 2018;St Louis et al., 2019). Global long-range atmospheric transport and deposition is the main pathway of Hg input to remote ecosystems (Lin et al., 2019;Ly Sy Phu et al., 2019;Sun et al., 2019). Soils account for more than 90% of Hg stored in terrestrial ecosystems (Obrist, 2012), with global top soil Hg pools (0–40 cm) estimated at > 300 000 Mg (Hararuk et al., 2013;Zhou et al., 2017a). The large soil Hg pools not only stem from geologic sources, but also from a legacy of historically anthropogenic emissions over the centuries (Obrist et al., 2014;Du et al., 2019).

Although many studies have focused on primary anthropogenic Hg emissions, releases from natural source materials is also an important pathway but with greater uncertainty and variability, including emissions from natural reservoirs (e.g. volcanic activity, geothermal sources, weathering of Hg from soil minerals) and re-emissions of previous deposited Hg. These natural sources can be equal to or two-fold larger than anthropogenic sources (Outridge et al., 2018;Fraser et al., 2018). Recent global Hg models estimate that 3600 Mg $yr^{-1}$ of atmospheric Hg is deposited to terrestrial surfaces, with 1000 Mg $yr^{-1}$ re-emitted back to the atmosphere (Outridge et al., 2018). Moreover, compared to primary anthropogenic emissions of Hg (2500 Mg $yr^{-1}$), estimates of re-emissions from soil surfaces are highly uncertain (Outridge et al., 2018;Wang et al., 2018). Compiling data from 132 studies, Agnan et al. (2016) found that the Earth's surface (particularly in East Asia) is an increasingly important source of total gaseous Hg (TGM) emissions, with up to half of the global emissions derived from natural sources. They estimated terrestrial TGM emissions of 607 Mg $yr^{-1}$, but with a large uncertainty range of −513 to 1353 Mg $yr^{-1}$. Additionally, a recent review suggests that future research should focus on campaigns to understand forest Hg behavior and long-term Hg observations, particularly in Asia (Zhang et al., 2019b).

Forest soils receive Hg inputs from: 1) throughfall that include wet deposition plus the wash off of Hg (II) deposited on foliage surfaces; 2) litterfall that contains foliage and other plant materials that have assimilated atmospheric Hg(0); and 3) direct dry Hg deposition to soil from the atmosphere

(Teixeira et al., 2018;Risch et al., 2017;Olson et al., 2018;Cheng et al., 2020). Mercury outputs from
forest soils occur from surface or subsurface runoff and air-land surface evasion. Forest soils are
highly complex media, with important features that affect soil-air exchange, including soil physio-
chemical characteristics (e.g., porosity, oxygen availability, redox potential, organic matter, pH)
(Obrist et al., 2010;Carpi et al., 2014). Other factors also influence this process, such as
meteorological conditions (e.g., solar radiation, air temperature, precipitation) (Zhou et al.,
2015;O'Connor et al., 2019), atmospheric chemistry (ozone, nitrate and hydroxyl radicals) (Peleg et
al., 2015;Angot et al., 2016), atmospheric TGM concentrations (Wang et al., 2007) and biological
processes (Obrist et al., 2010;Chen et al., 2017). Therefore, to characterize and quantify land-
atmosphere exchange of TGM, it is necessary to understand the roles of these factors in mediating
this process.
Field studies have shown that elevated anthropogenic Hg emissions in South-East Asia have
resulted high atmospheric Hg concentrations an deposition regionally (Kumari et al., 2015;Pan et
al., 2010;Zhang et al., 2019b). Forests experience particularly elevated net Hg loads due to enhanced
deposition associated with the tree canopy, especially in China (Wang et al., 2016;Zhang et al.,
2019a). The annual loading of THg to subtropical forests in China have been shown to be much
higher than forest catchments in Europe and North America (Wright et al., 2016;Zhou et al., 2020).
High Hg deposition has resulted in elevated soil Hg pools in Chinese subtropical forests (Wang et
al., 2018;Wang et al., 2009). In contrast, a recent study showed that the Hg deposition and soil Hg
concentrations at a temperate forest in China were similar to those in Europe and North America
(Zhou et al., 2020). The forested area in China is $2.2 \times 10^4$ km$^2$, with about 50% and 40% occurring
in subtropical and temperate zones, respectively. Therefore, it seems likely that subtropical and
temperate forests in China, with contrasting climate, vegetation cover, and atmospheric Hg
deposition, may also show different patterns of Hg cycling.
Forest ecosystems not only act as sinks for atmospheric Hg deposition, but can also serve as
sources resulting from legacy Hg that has accumulated in surface soil. For example, one study
constructed the Hg budget in subtropical forest in southern China showing that the forest is a minor
sink for atmospheric Hg but a significant net Hg(0) source (58.5 $\mu$g m$^{-2}$ yr$^{-1}$) (Yu et al., 2020). In
contrast, another study also in southern China using budgets of air-foliage and air-soil Hg(0)
exchange fluxes, showed that forest is a net sink of Hg(0) (20.1 $\mu$g m$^{-2}$ yr$^{-1}$) (Yuan et al., 2019a;Yuan
et al., 2019b). These results indicate that there is considerable uncertainty and variability in the
source-sink behavior of Hg in subtropical forests of southern China. Furthermore, no studies have
conducted in northern China to characterize the Hg fluxes in the temperate forest.
There has been much research characterizing Hg fluxes between the forest floor and the
atmosphere from studies worldwide, as reviewed by Zhu et al. (2016) and Agnan et al. (2016). In
this paper, we present measurements on atmosphere-land Hg fluxes conducted over 130-days and
96-days, respectively, during four seasons for five sites both at a temperate forest catchment at Mt.
Dongling (MDL) and a subtropical forest catchment at Tieshanping Forest Park (TFP) in China. The
aims of this investigation were to (1) characterize the air-land surface Hg fluxes in different
terrestrial ecosystems; (2) conduct detailed field measurements to characterize the uncertainty of
land use and climate change in air-surface fluxes of TGM in forest catchments; and (3) to compare
estimates of Hg emissions from forest soils at temperate and subtropical zones. We hypothesize that
a multi-plot and multi-seasonal study of soil-air fluxes in each forest system will provide new
perspectives on the climate change and land use on the soil-air Hg fluxes, and improve
understanding and estimates of soil Hg evasion from forest ecosystems.

**2. Materials and methods**
**2.1. Study area**
This study was conducted at TFP in the subtropical zone (106°41.24′E, 29°37.42′N) and at
MDL in the temperate zone (115°26′, E40°00' N) in China (Fig. 1). The TFP is dominated by a
Masson pine (*Pinus massoniana* Lamb.) stand (conifer) with some associated species, such as
camphor (*Cinnamom camphora*) and Gugertree (*Schima superba* Gardn), which were planted in
1960s following the loss of a natural Masson pine forest. The forest is located about 20 km northeast
of Chongqing City, at an altitude from 200 to 550 m. The mean annual precipitation is 1028 mm,
with 75% of the rainfall occurring from May to October. The mean annual air temperature is 18.2 ℃.
The total area of the study forest in the TFP is $1.06×10^3$ ha. The soil is typically mountain yellow
earth (corresponding to an Acrisol in the FAO) (FAO, 1988), with clay mineralogy dominated by
kaolinite (Zhou et al., 2016).
Mt. Dongling is near the Beijing Forest Ecosystem Research Station, Chinese Academy of
Sciences, which is located 110 km southwest of mega-city Beijing in North China. The elevation is
1300 m asl. The annual average rainfall is 612 mm and mean relative humidity is 66%. The climate
of the region is predominantly warm temperate continent monsoon with an annual average
temperature of 4.8 °C and precipitation of 611.9 mm. Soil type is mountain brown earth
(corresponding to a Eutric cambisol in FAO) (FAO, 1988) (Zhou et al., 2018). The relatively cool
climate in the study area has resulted in deep litter and high organic matter concentrations (Fang et
al., 2007). The study area is a mature, secondary forest protected since the 1950s following the
extensive deforestation. Hg concentrations in environmental media at the site are provided in the
Supporting Information (SI, Supporting Text).

**2.2. Dynamic flux chamber (DFC) measurement**
To reduce the spatial uncertainty in Hg fluxes, different ecosystems were selected for study in
a sub-catchment at the subtropical TFP, including a coniferous forest (plots S-A and S-B), a wetland
(plot S-C), a broad-leaved (camphor) forest (plot S-D) and an open field with bare soil (plot S-E),
and a sub-catchment at the temperate MDL, including a Chinese pine forest (plot T-A), larch forest
(plots T-B), wetland (plots T-C), mixed broad-leaved forest (plots T-D) and open field (plots T-E)
(Fig. 1). To reduce temporal uncertainty in Hg fluxes, 130-days and 96-days of flux observations
were undertaken over four seasons (about one-month of continuous observations for each season,
except for one-week during winter at the MDL) (Table S1). The locations of each plot is described
in the Table 1 and illustrated in Fig. 1.
Semi-cylindrical quartz glass and open-bottom DFCs (4.71 L) were utilized during the
sampling campaign. The area of the DFCs over the soil surface was $20 \times 30$ cm, with six inlet holes
(1 cm diameter) (Fig. S1). Local fine grained soil was placed outside the chamber to seal any gap
between the base of the chamber and the soil. At the outlet of the chamber, an orifice was connected
to two exit tubes: one to a regulated suction pump and the other to a gold cartridge for trapping
outlet TGM. A sub-stream of air was trapped by a pair of gold quartz cartridges at a flow rate of 0.5
L min$^{-1}$, which was measured using an integrating volume flow meter. All the gold cartridges were
constructed with gold silk (< 0.5 mm diameter). The strands of gold silk were rolled together in a
small coil and about 15 coils were used to fill a quartz cartridge with about 2 g of gold. The accuracy
of all traps were evaluated (see section 2.4) and non-conforming cartridges were discarded. The
chamber flushing flow turnover time (TOT) was 0.47 min and 0.94 min for the subtropical forest

and temperate forest plots, respectively. The Hg flux was calculated using the following equation:

$$F = (C_0 - C_i) \times Q/A$$

where $F$ is the soil Hg flux (ng m$^{-2}$ hr$^{-1}$); $C_o$ and $C_i$ are the steady state Hg concentrations (ng m$^{-3}$) of the outlet and inlet air streams, respectively, which were calculated by the Hg mass detected in gold cartridges and the corresponding air volume; $A$ is the surface area enclosed by the DFC; $Q$ is the flow rate of ambient air circulated through the DFC (10 L min$^{-1}$ for TFP and 5 L min$^{-1}$ for MDL).

High flow rates and short TOTs are appropriate for measuring flux from soils with high Hg concentrations or emissions, while lower flow rates and longer TOT are more appropriate for soils with low Hg concentrations or emissions. Eckley et al. (2010) suggested that the optimal flow was at the beginning of the stable $C_0 - C_i$ ($\Delta$C) period, which was chosen as a compromise between competing criteria aimed at creating conditions inside the DFC similar to the adjacent outside air. Our previous study showed that when $\Delta$C was relative stable, the corresponding flushing flow rate was from 5 to 10 L min$^{-1}$ at the subtropical forest (Zhou et al., 2017a). To avoid suppression of Hg emissions due to the excessive buildup of Hg within the chamber, the flow rate of ambient air circulated through the DFC was 10 L min$^{-1}$ at the subtropical forest. At the temperate forest, the soil Hg concentrations was about 3-4 times lower than those at the subtropical forest, so the lower flow rate of 5 L min$^{-1}$ was used at these plots. The DFC chambers in all plots were moved every week to mitigate against changes in soil moisture due the covering of soil by the chambers. If a precipitation event occurred, the chambers were also moved to new positions during the sampling period (morning or evening) to be representative of soil conditions receiving ambient precipitation.

The pair of gold cartridges for each DFC were collected twice a day: every morning (about 8:00) and afternoon (about 17:00) representing night (17:00−8:00 of next day) and day (8:00−17:00) emissions, respectively. Twenty gold quartz cartridges were alternated during the sampling program. In addition, diurnal variations of soil-air Hg fluxes were also conducted in each season, with gold cartridges collected every half an hour. A total of four diurnal measurements were conducted over the study in each forest, with diurnal variations were measured one day per season. It has been reported that the DFC measurements can introduce bias under a given design flushing air flow rates and environmental condition (Lin et al., 2010;Zhang et al., 2002). The DFC enclosure imposes a physical constraint that can lead to accumulation in or evasion from the soil surface under measurement. Extensive experiments were conducted at our plot sites to determine the appropriate

experimental conditions for accurate measurements. We followed recommendations made by Eckley et al. (2010) for our measurements.

### 2.3. Environmental measurements

At each sampling plot, soil samples were collected from the DFC footprint (0−5 cm). Soil Hg and soil organic matter (SOM) concentrations were measured using a DMA-80 direct Hg analyzer (Milestone Ltd., Italy) and loss on ignition (LOI) method, respectively, using methods detailed in the SI. Soil percent moisture and temperature were monitored with Time Domain Reflectometry (TDR) Hydra Probe II (SDI−12/RS485) and a Stevens water cable tester (USA). Solar radiation was measured by a weather station (Davis Wireless Vantage VUE 06250 Weather Station, Davis Instruments, Hayward, CA) located in the TFP Forest Station and Beijing Forest Ecosystem Research Station, within about 500 m of each plot.

### 2.4. Quality assurance and quality control (QA/QC)

All cartridges were transported to a laboratory at the TFP Forest Station for Hg determination using a cold vapor atomic fluorescence spectroscopy (CVAFS) detector (Brooks Rand III). The limit of detection, based on three times the standard deviation of replicate measurements of the blank was 1 pg. Based on the sampled air volume, the detection limits were < 0.10 ng m$^{-3}$. A calibration curve was developed using Hg saturated air and the calibration curve was required to have a correlation coefficient greater than 0.99 before the samples analysis could proceed. Before and after the measurement of the sampling cartridges in each day, standard Hg saturated air was injected to test the accuracy of the Hg analyzer. If the deviation of the measured Hg mass was higher than 5%, new calibration curve would be developed.

A controlled volume of saturated Hg air at a known temperature was injected to measure Hg recovery from the gold cartridges before and after the campaigns in each season. The recoveries of gold cartridges before and after the operation ranged from 98.8 to 103.2% and 96.3 to 102.5% (n=155, average=98.9%), respectively. The collection efficiency of Hg vapor by the gold cartridges was determined by connecting two cartridges in sequence and sampling the ambient air over 24 h in the laboratory. For all cartridges, less than 1% Hg was detected on the second cartridges compared to the first cartridge, indicating that more than > 99% of TGM was absorbed by the gold cartridges

during the field operation. For comparison, Hg fluxes were measured by two chambers side by side
simultaneously. Blanks of the soil TGM flux sampling systems were measured by placing the DFC
on a quartz glass surface in the five plots. The sampling time for blank measurements was same as
soil-air TGM flux measurements, which were collected at 8:00 and 17:00, representing night
(17:00−8:00 of next day) and day (8:00−17:00) emissions, respectively. The averaged blank was
$0.13 \pm 0.21$ ng m$^{-2}$ h$^{-1}$ (n=10), which was subtracted from the soil-air TGM flux for each season.

**2.5. Statistical analysis**
Structural equation modeling (SEM) were performed on the collected Hg flux data using Amos
software. SEM, developed from a fully conceptual model using $\chi 2$ tests with maximum likelihood
estimation, was conducted to infer the interplay of temperature, solar radiation, soil moisture, and
air TGM concentrations on measurements of soil-air TGM exchange fluxes. Seasonal and annual
fluxes were compared among the ten plots. Separate two-way ANOVAs were used to determine if
differences in Hg fluxes existed among the seasons and sites. All differences in mean values were
significant at the p=0.05 level and all means are reported with $\pm$ one standard deviation from the
mean. The correlations between environmental parameters and fluxes were analyzed by Pearson's
Correlation Tests using SPSS software (SPSS Inc. 16.0) and correlation coefficient and p values are
presented and significantly correlated at the level of 0.05.

**3. Results and discussion**
**3.1. Landscape- and forest species-dependence of soil-air Hg fluxes at the forest catchment**
**scale**
The soil TGM flux measurements for the five plots were calculated for the day and night and
reported as mean daily fluxes with standard deviations (SD) at the subtropical (Fig. 2a) and
temperate forests (Fig. 2b). Over the course of the campaigns, net TGM emission was observed at
the open field ($24 \pm 33$ ng m$^{-2}$ hr$^{-1}$), coniferous forest (upper elevation $2.8 \pm 3.9$ ng m$^{-2}$ hr$^{-1}$, mid
elevation $3.5 \pm 4.2$ ng m$^{-2}$ hr$^{-1}$) and the broad-leaved forest ($0.18 \pm 4.3$ ng m$^{-2}$ hr$^{-1}$), while net
deposition was evident at the wetland ($-0.80 \pm 5.1$ ng m$^{-2}$ hr$^{-1}$), respectively, at the subtropical
forest. At the temperate forest, net TGM emission was observed at the wetland ($3.81 \pm 0.52$ ng m$^{-2}$
hr$^{-1}$), open field ($1.82 \pm 0.79$ ng m$^{-2}$ hr$^{-1}$), mixed broad-leaved forest ($0.68 \pm 1.01$ ng m$^{-2}$ hr$^{-1}$),
larch forest ($0.32 \pm 0.96$ ng m$^{-2}$ hr$^{-1}$), while net deposition was evident at the Chinese pine forest
($-0.04 \pm 0.81$ ng m$^{-2}$ h$^{-1}$), respectively. The fluxes at the temperate forest were 10-times lower than
values at the subtropical forest due to different environmental factors, such as lower temperature,
solar radiation and soil Hg concentrations (see section 3.3).
These patterns suggest that soil-air Hg fluxes at catchment scale vary by soil properties (e.g.,
soil Hg concentration, moisture, SOM) and forest species composition. High variability, as
evidenced by high SD and coefficient of variation (SD/mean, range of 14−2374%), was evident in
daily Hg fluxes largely driven by meteorological variation. The fluxes at the subtropical forest
plots of this study were much lower than those reported for other subtropical evergreen forests in
China such as Mt. Gongga (0.5–9.3 ng m$^{-2}$ hr$^{-1}$) (Fu et al., 2008), Mt. Jinyun (14.2 ng m$^{-2}$ hr$^{-1}$)
(Ma et al., 2013) and Mt. Simian (11.23 ng m$^{-2}$ hr$^{-1}$) (Ma et al., 2018), all of which were generally
conducted during sunny days. Our flux measurements at temperate forest were slightly lower or
comparable to those in North American deciduous forests, ranging from −0.73 to 2.7 ng m$^{-2}$ hr$^{-1}$
(Choi and Holsen, 2009b;Hartman et al., 2009;Carpi et al., 2014;Ma et al., 2018). These results
demonstrated that measurements over several days may exhibit considerable temporal variability
and long-term study should be undertaken to reduce the uncertainty in temporal patterns of soil Hg
emissions.
The mean TGM fluxes in the open fields were about 10 and 6 times higher than those under
the forest canopy at the subtropical and temperate forests, respectively ($p < 0.001$). Our results are
consistent with Ma et al. (2013) and Xin and Gustin (2007), showing large Hg evasion following
forest conversion to bare soils due to direct exposure to sunlight, as fluxes were enhanced by
increases in solar radiation and temperature. Due to frequent heavy rains at the subtropical forest
catchment, a large amount of surface runoff impacted the wetland (plot S-C). Elevated runoff may
have decreased Hg ($96 \pm 43$ ng g$^{-1}$) and SOM in surface soils due to erosion (Table 1). This site had
the lowest TGM fluxes of the plots studied at the subtropical forest (overall net sink). In addition,
soils in the wetland plot were mostly saturated throughout the year, limiting Hg fluxes and likely
contributing to the sink behavior. In contrast, the mean annual rainfall was 40% lower at the
temperate forest and the wetland was located at relatively lower terrain. Litter from surrounding
higher terrain forest accumulated in the low lying wetland. The cool and dry climate also contributed
to high organic matter and low bulk density (Fang et al., 2007). Higher SOM likely facilitated
binding of trace metals, leading to high soil Hg concentrations (117 ng g$^{-1}$) at the temperate wetland.
These conditions were conducive to biological activity, promoting the mineralization of SOM and
the release of volatile Hg(0) from soil (Choi and Holsen, 2009b;Osterwalder et al., 2019). The
wetland had the highest TGM fluxes of the plots studied at the temperate forest (overall net source).
Previous studies have suggested that soil water is able to mobilize Hg from binding sites on soil
(Gustin, 2003;Kocman and Horvat, 2010) and high soil water decreases soil redox potential (Zarate-
Valdez et al., 2006), both of which can facilitate the conversion of Hg(II) to Hg(0). Additionally, the
climate is relatively dry in north China, especially in spring. The high solar radiation and relatively
high air temperature not only enhance the reduction of Hg(II) to Hg(0), but also increase water
evaporation compared to other study sites. Enhanced water evaporation at higher temperature,
facilitates Hg emissions from soils (Gustin and Stamenkovic, 2005;Lin et al., 2010). Additionally,
given that Hg conversion to Hg(0) in soil profiles occurs mainly via biotic processes, maximum
aerobic microbial activity has been delineated at soil water content equivalent to 60% of a soil's
water holding capacity (Breuer et al., 2002;Kiese and Butterbach-Bahl, 2002). Appropriate soil
moisture in the wetland would likely enhance the microbial reduction of Hg(II) to Hg(0). Therefore,
the highest Hg flux was observed in the temperate wetland, especially in spring. The main reasons
for the significant differences between the soil Hg fluxes at the two wetland sites is likely that the
saturated soil at the subtropical forest inhibited Hg(0) evasion (Gustin and Stamenkovic, 2005) (see
section 3.3).
At the subtropical forest, litterfall in the broad-leaved (camphor) plot (plot S-D) was twice as
high as that of the coniferous (pine) plot (plots S-A and S-B) (Zhou et al., 2018), likely resulting in
greater shielding of sunlight to the surface soil and limiting soil Hg evasion. Increases in light
transmission through the canopy increase both solar radiation and soil temperature, which can
enhance photochemical reduction of Hg(II) at the soil surface and Hg(0) evasion. In the mid-slope
of the pine stand (plot S-B), soil Hg concentration was elevated compared to the upslope plot (Table
1), with corresponding with higher soil Hg fluxes. At the temperate forest, the lowest Hg flux and
overall deposition was observed at the evergreen forest of Chinese pine, where the canopy cover
likely limited Hg flux by decreasing solar radiation to soil and warming. Similar at the subtropical
forest, the needle biomass in the larch plot was about 2.5 times greater as that in the mixed broad-
leaved plot (plot T-D) at the temperate forest, resulting in shielding the sunlight to the surface soil
and limiting soil Hg evasion at larch plot.

The forest canopy not only influences the soil Hg concentration by mediating atmospheric Hg

deposition (Zhou et al., 2018;Zhou et al., 2017a), but also alters soil physio-chemical properties (e.g.
SOM, pH, porosity) (Mo et al., 2011) and microbial communities (Nagati et al., 2020), which affect
soil-air exchange. For example, the annual litterfall Hg deposition flux at the broad-leaved plot (91
$\mu g\ m^{-2}\ yr^{-1}$) at the subtropical forest was approximately two times greater than the coniferous plot
(41 $\mu g\ m^{-2}\ yr^{-1}$) (Zhou et al., 2018). Conversely, the SOM and soil Hg concentrations in the broad-
leaved forest were lower than the coniferous forest. Moreover, litter decomposition rate was lower,
but the Hg mass accumulation in the litter was much higher in the coniferous forest compared to the
broad-leaved forest due to higher throughfall Hg deposition at the coniferous plot (Zhou et al., 2018),
which resulted in a seemingly inconsistent pattern between litterfall mass and SOM, as well as
litterfall Hg deposition and soil Hg concentrations. At the temperate forest, the higher litterfall Hg
deposition and lower litter decomposition in the larch plot compared to the broad-leaved plot (Zhou
et al., 2017a), resulted in significant higher SOM and soil Hg concentrations (Table 1). Tree species
can change the physicochemical properties of soil (e.g. SOM, soil Hg concentrations) and influence
soil-air exchange. These biological factors likely contribute to the much lower TGM evasion in the
broad-leaved plot than the coniferous plot at the subtropical forest, but much higher TGM evasion
in the broad-leaved plot than the deciduous needle (larch) plot at the temperate forest (Fig. 2).

Most studies measure soil TGM fluxes at only one location or at a single forest stand to

characterize the whole ecosystem. Our observations clearly show that soil-air Hg fluxes vary
substantially across different plots (Fig. 2), indicating that forest type/cover and landscape position
significantly affect the TGM fluxes and therefore the spatial variability in soil Hg fluxes among
different sub-plots must be considered. Based on the areal distribution of each plot type in the study
sub-catchments of the subtropical forest (coniferous upland and mid-slope, broad-leaved, wetland,
open) (4.6 ha) and the temperate forest (Chinese pine, larch, wetland, mixed broad-leaved and
open) (5.0 ha) (Table S1), the area-weighted TGM flux was 3.2 and 0.32 ng $m^{-2}\ hr^{-1}$ for the entire
subtropical and temperate catchments, respectively. The area-weighted TGM fluxes were 14%
higher than plot S-A and 16% lower than plot S-B of the Masson pine stand at the subtropical
forest, and were 907% higher than Chinese pine plot and 53% lower than mixed broad-leaved plot
at the temperate forest, respectively. The observations at several plots with diverse forest cover in
this study should reduce the overall uncertainty associated with soil-air fluxes of TGM in the
overall forest catchment.

**3.2. Seasonal variations of soil-air Hg fluxes at the forest catchment scale**
Soil TGM fluxes not only exhibited clear seasonal variations at all the plots, but also were
responsive to phenological and meteorological patterns. At the subtropical forest, soil Hg fluxes
were generally highest in the summer (Fig. 2a), which showed net emissions at all the five plots,
followed by spring and autumn, with the lowest values during winter, which exhibited net deposition
at all plots with the exception of plot S-B. The observed seasonal variation was dependent on
sunlight (Fig. 3), because solar radiation drives photochemical reduction of Hg(II) (note the
correlation between the TGM fluxes and solar radiation, Fig. S2). Additionally, greater solar
radiation increases temperature, which promotes the production of soil Hg gas by biological and
abiotic processes. At the temperate forest, the Hg fluxes were the highest in the deciduous forest
plot (wetland, mixed broad-leaved forest and larch forest) in spring before leaf-out when solar
radiation could directly reach the forest floor (Fig. S3). In the open field and evergreen forest
(Chinese pine forest) plots, the Hg fluxes were highest in summer with the highest solar radiation
and temperature (Fig. 4 and Fig. S3). The lowest Hg fluxes were measured in the winter at all plots
when the soil was covered with snow, with net Hg emission observed at the open field and net
deposition observed at the other four sites (Fig. 2b).
We also observed strong variation in TGM evasion under different weather conditions. Rain
events decreased TGM fluxes at all plots in both forests (Fig. S4), as the rainwater decreased soil
pore space leading to decreases in evasion from soil. Furthermore, the solar radiation and
temperature during rainy days was much lower than those for sunny days for a given season (Fig. 3
and Fig. 4). Manca et al. (2013) studied snow-air Hg exchange at Ny-Ålesund, showing on average
a small net deposition -0.24 ng $m^{-2}$ $hr^{-1}$. Likewise, overall deposition between -0.6 and -23.8 ng $m^{-2}$
$hr^{-1}$ were observed at snow-covered agricultural areas at Northeastern China (Wang et al.,
2013;Zhang et al., 2013). However, some studies of snowpack have shown net Hg deposition at
nighttime and net emissions during daytime due to high solar radiation (Maxwell et al.,
2013;Spolaor et al., 2019). Empirical models suggest that most of the Hg(0) deposited to snow was
re-emitted back to the atmosphere (Durnford and Dastoor, 2011). During the campaigns in winter,

the solar radiation was relatively lower, which may be why net deposition occurred (Fig. 4). Additionally, refrozen ice/snow layers are characterized by elevated Hg concentrations and the deposited Hg from atmosphere could be potentially released to meltwater (Zhang et al., 2012;Perez-Rodriguez et al., 2019), which is consistent with our results that atmospheric Hg deposition could release to meltwater during snow melt. Our observations through the annual climatic cycle reduce uncertainty and bias of temporal patterns of soil-air Hg fluxes. Moreover, multi-plot observations reduce the uncertainty and bias associated with spatial variation. Together these more detailed measurements improved estimates of overall ecosystem soil Hg evasion, and confirm our hypothesis.

### 3.3. Correlations between environmental factors and fluxes

To investigate the correlation between soil-atmosphere fluxes and environmental factors, data over the four seasons were used. These data offer a long continuous time series for the five measurement plots in each forest (Fig. 3 and 4). According to a global database, atmospheric fluxes at Hg-enriched sites are positively correlated with substrate Hg concentrations, but this relationship is not observed at sites with lower background concentrations of soil Hg (Agnan et al., 2016). Our soil Hg fluxes were strongly correlated with soil Hg concentrations at vegetated sites (forests and wetland) at the subtropical forest (Fig. S5), but not at the temperate forest.

Photo-reduction is a major driver of TGM evasion from the Earth's surface (Howard and Edwards, 2018;Kuss et al., 2018;Gao et al., 2020). This process is due to photochemically mediated reduction that converts soil water Hg(II) to volatile Hg(0) and enhances the Hg(0) pool in soil pores (Xin and Gustin, 2007;Choi and Holsen, 2009a). Therefore, the elevated soil pore Hg(0) concentrations increased the potential for TGM diffusion from soil to the atmosphere, which drives an increase of Hg emissions from soil. At all the study sites no matter the daily average fluxes (Fig. 3 and 4), daytime fluxes (Fig. S2 and S3) were all significantly correlated with solar radiation, and the solar radiation also increased daytime fluxes compare to nighttime values (Fig. S6). In the evergreen plots of the subtropical (plots S-A, S-B, S-D) and temperate (plot T-A) forests, Hg fluxes were the most highly dependent on soil temperature compared to the solar radiation during the four seasons, likely due to evergreen canopy limiting solar radiation to the forest floor. With the consistent shade of the coniferous forest canopy, the Hg flux was highly dependent on soil surface temperature rather than solar radiation to the forest floor.

To consider synergistic effects from multiple factors, SEM was applied to infer the soil-air
TGM exchange processes (Fig. 5). It is clear that temperature was a more dominant factor driving
air-soil TGM exchange flux over the four seasons in the subtropical forest plots, while solar
radiation was a more dominant factor at the temperate forest due to direct exposure of the forest
floor to solar radiation the leaf-off seasons. At the open fields of both forests, temperature and solar
radiation had a synergistic effect on soil Hg fluxes. A recent study of soil-air TGM fluxes at
subtropical evergreen broadleaf forest in South China also suggested that temperature is the most
important driver of air-soil TGM exchange (Yuan et al., 2019b). Therefore, we may infer that under
the shade of the forest canopy, temperature is the dominant factor causing variation in TGM evasion
from forest soil.
Mercury fluxes in wetlands in both forests (plots S-C and T-C) were less strongly correlated
with soil temperature compared to the other plots in both forests (Fig. S7 and S8). Generally,
temperature is an important factor that promotes TGM evasion after its formation from Hg(II) more
by biotic than abiotic processes in soils (Pannu et al., 2014). The Hg(0) in soil pore gas mainly
results from biotic production. For example, soil sterilization can decrease Hg converted to Hg(0)
by ~50% ; additionally, 1% of the soil Hg is converted to Hg(0) via abiotic processes, compared to
6.8% by biotic processes at 283 K, and the fraction of Hg reduction by biotic processes increases
with temperature increases (Pannu et al., 2014). At the subtropical forest, the wetland soil was
largely saturated. This condition likely limited soil pore release of TGM to the atmosphere, resulting
in a weaker correlation between soil temperature and Hg fluxes. Furthermore, the Hg exchange
fluxes were more dependent on solar radiation and less dependent on temperature during the leaf-
off period at the temperate deciduous plots; therefore, the Hg fluxes were more solar radiation-
driven in the deciduous forests, especially in the wetland (Fig. S3 and S7).
During the campaign, significant negative correlations were evident between soil moisture and
soil-air fluxes of TGM at the five plots at the subtropical forest ($r^2 = 0.03-0.39$, $p < 0.05$ for all, Fig.
S9), but there was no significant correlations with soil moisture for the temperate forest (Fig. S10).
Generally there is an optimum soil moisture condition that maximizes soil TGM flux (Gustin and
Stamenkovic, 2005;Lin et al., 2010;Obrist et al., 2014;Osterwalder et al., 2018;Johnson et al., 2003),
which ranges from 60% to 80% of the water holding capacity of a soil (Pannu et al., 2014). A
laboratory experiment using undisturbed soil collected from the our subtropical study area showed

that increasing soil moisture from 2% to 20% increased the TGM flux 80% at 24 °C (Wang et al., 2014). A second field experiment was conducted to study the effects of higher soil moisture on TGM flux at the subtropical forest, showing that increasing soil moisture gradually decreased the soil Hg emissions over the range of 31−39% (Zhou et al., 2017b). Combining the results of these experiments, the soil Hg fluxes at the subtropical forest catchment appear to increase from low values of soil moisture reaching an optimum in the range of 20-30% and then decreasing with increasing soil moisture above these values. In the current study, we observed following an extended dry period with an extended wet period enhanced the Hg fluxes in both forests; however, individual rainfall events did not enhance or decrease the Hg fluxes due to short-term increases in soil moisture and lower solar radiation associated with those events (Fig. 3 and 4). Additionally, Lin et al. (2010) observed the synergistic effects (20–30 % of additional flux enhancement) between air temperature (15 and 30 °C) and soil moisture (2.5 and 27.5 %). Perennially humid weather results in relatively high soil moisture at the subtropical forest (largely > 25% during the campaigns). Considering the relatively high bulk density and low porosity of soil at the subtropical forest (Sørbotten, 2011), soil moisture likely exceeded the optimum range for TGM evasion during the campaigns resulting in significantly negative correlations (Fig. S9). In contrast, lower bulk density and higher soil porosity would result in higher optimum range of soil moisture at the temperate forest. Moreover, the temperate forest had a large range of soil moisture (2 to 60%) in the five plots which combined with the synergistic effects of soil moisture with temperature (Lin et al., 2010), resulted in a condition where moisture was not a main driver of TGM evasion.

Soil-air Hg fluxes also showed significant negative correlations with atmospheric TGM concentrations at the ten plots at both forests ($r^2 = 0.023−0.26$, $p < 0.05$, Fig. 6 and 7), which had a greater effect than soil moisture at both forests, except for plots T-C, S-A and S-E (Fig. 5). According to the two-resistance exchange interface model, the exchange fluxes of Hg are controlled by the gradient of TGM concentrations at both interfaces (Zhang et al., 2002). As a result elevated atmospheric TGM concentrations should decrease the diffusion of soil pore TGM to the atmosphere. In a companion study, the soil pore TGM concentrations were measured at all the plots at the subtropical and temperate forests, except the wetlands (Zhou et al., in review). These results showed that gradient of TGM concentrations between the surface air and pore air at 3 cm were significantly correlated with the soil-air TGM fluxes at all the plots (Fig. S11 and S12). These results are

consistent with an experiment conducted at this subtropical forest, where artificially increasing
ambient-air TGM concentrations significantly inhibited soil Hg volatilization (Zhou et al., 2017b).
SEM inferred that that air TGM concentrations was the second important driver influencing the soil-
air TGM exchange in Masson pine (Plot S-B), evergreen broad-leaved and wetland plots at
subtropical forest (Fig. 5).

Xin and Gustin (2007) and Gustin et al. (2006) defined an associated concept of the

compensation point for soils, which is the atmospheric Hg concentration at which the net Hg flux
between the soil and the atmosphere was zero. If the atmospheric TGM concentration is above
compensation point, atmospheric deposition occurs; if the concentration is below the compensation
point soil emission occurs. A strong linear relationships are shown in Figs. 5 and 6 ($p < 0.01$),
resulting in compensation points of 2.47, 2.97, 6.00, 3.33 and 3.50 ng m$^{-3}$ for Chinese pine, larch,
wetland, mixed broad-leaved forests and open field at the temperate forest with area-weighted
compensation point of 3.42 ng m$^{-3}$. The compensation points were much higher at the subtropical
forest, with values of 6.50, 7.71, 3.92, 3.83 and 12.91 ng m$^{-3}$ for Masson pine upland and mid-slope,
wetland, broad-leaved and open field at the subtropical forest with area-weighted compensation
point of 6.82 ng m$^{-3}$. Another study of subtropical coniferous forest showed similar compensation
point (7.75 ng m$^{-3}$) as those in the Masson pine forests of our study (Luo, 2015).

Diurnal variation in soil-air TGM fluxes were measured in plot S-A at the subtropical forest

(Fig. 8) and in plot T-D at the temperate forest (Fig. 9). Soil TGM fluxes were well correlated with
soil and air temperature ($p < 0.01$ for all) and were highly dependent on solar radiation in spring,
summer and autumn ($p < 0.01$ for all) but not in winter ($p > 0.05$), which are similar to seasonal
patterns from other studies (Howard and Edwards, 2018;Osterwalder et al., 2018;Johnson et al.,
2003). Solar radiation has been shown to promote photochemical reduction of soil-bound Hg and
enrich Hg(0) in soil pore gas. This reaction is kinetically enhanced at higher temperatures (Eckley
et al., 2015;Lin et al., 2010;Zhang et al., 2001). Compared to the other three seasons, the relatively
low soil temperature (5.95 °C at the subtropical forest and –5.66 °C at the temperate forest) may
have limited the relationship between soil TGM flux and solar radiation during the winter season.

**4. Conclusions and study implications**

Prior to undertaking these measurements of Hg air-surface exchange flux, no direct

measurement of Hg exchange flux were available for background landscapes in North China. Our
detailed direct observations have important implications for the role of forests in global and regional
Hg cycles. Through multi-plot measurements over 130 and 96 days at the subtropical and temperate
forests in China, we were able to reduce the uncertainty of soil-atmosphere TGM fluxes at the
catchment scale and improve understanding of how landscape attributes contribute to variability in
soil Hg evasion. It is inferred that forest soils acts as net TGM sources to the atmosphere. Strong
correlations were evident between the soil Hg flux and environmental variables in some plots, such
as solar radiation, temperature, soil moisture and air TGM concentrations.
The compensation points were determined for background forest soils from full-scale field data
showing area-weighted values of 6.82 and 3.42 ng m$^{-3}$ for the entire subtropical and temperate
catchments, respectively. The values of compensation indicate that the atmospheric TGM
concentration can play a critical role in limiting TGM fluxes between forest floor and atmosphere.
Future studies need to focus on forest soils as an important increasing source of Hg to the
atmosphere, because of recent declines in anthropogenic Hg emissions and TGM concentrations
(Liu et al., 2019). Moreover TGM re-emissions are partially derived from legacy Hg stored in
surface soils. A recent study using models simulating the dynamics of the subtropical forest
landscape under climate change, harvesting, and land-use disturbances in southern China showed
that coniferous forest area increased approximately 3.7 times compared to broad-leaved forest area
(Wu et al., 2019). In the temperate forest, climatic changes in the northern China are expected to
cause coniferous stands to transition to deciduous forests over the next hundred years (Ma et al.,
2014). Climate change and land-use disturbance may increase the compensation points in both
temperate and subtropical forests, therefore, increasing emissions of legacy Hg from terrestrial soils
to the atmosphere. Some studies have emphasized that climate and land use change will potentially
enhance deposition of Hg to forested landscapes (Haynes et al., 2017;Richardson and Friedland,
2015;Li et al., 2020); however, our study suggests that legacy Hg in forest soils could be emitted
back to atmosphere, offsetting enhanced atmospheric Hg deposition. Better understanding of the
response of Hg emissions from forest soils to climate and land use change is an important topic for
future research.

*Data availability*. The data will be available upon request to the corresponding author.

*Author contributions*. ZW and XZ conceived the experiment; JZ conducted the measurements; JZ wrote the paper with inputs from CTD, CL and ZW. All authors reviewed the manuscript.

*Competing interests.* The authors declare that they have no conflict of interest.

*Acknowledgements.* This work was funded by the Second Tibetan Plateau Scientific Expedition and Research Program (STEP, Grant No. 2019QZKK0307), the Natural Science Foundation of China (No.42077345 and No. 42077381),the National 973 Program of China (2013CB430002) and the National Key Research and Development Program of China (2017YFC0210106). The authors would like to thank Mingquan Zou and Beijing Forest Ecosystem Research Station, Chinese Academy of Sciences, for the help in our sampling and providing the meteorological data.

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

**Table 1.** Locations and summary of measurements (mean ± standard deviation) of soil-air TGM fluxes and environmental parameters at ten plots at the subtropical and
temperate forests.

| Forest | Plots | Locations | Flux (ng m$^{-2}$ hr$^{-1}$) | Soil surface TGM (ng m$^{-3}$) | Soil Hg concentration (ng g$^{-1}$) | SOM (0-5, %) | Soil moisture (%) | Soil temperature (°C) | Solar radiation (W m$^{-2}$) |
|---|---|---|---|---|---|---|---|---|---|
| Subtropical forest | Plot S-A | Top-slope of coniferous forest | 2.8 ± 3.9 | 3.6±1.3 | 219±15 | 13.6 | 0.3±0.1 | 16.8±7.6 | 39.9±27.5 |
| | Plot S-B | Middle-slope of coniferous forest | 3.5 ± 4.2 | 3.8±1.3 | 263±22 | 16.3 | 0.4±0.1 | 16.9±7.7 | 40.2±27.5 |
| | Plot S-C | Wetland | -0.80 ± 5.1 | 3.7±1.4 | 96±43 | 4.9 | 0.3±0.1 | 16.7±7.5 | 30.5±27.9 |
| | Plot S-D | Broad-leaved forest | 0.18 ± 4.3 | 3.3±1.4 | 156±17 | 8.8 | 0.3±0.1 | 16.9±7.6 | 20.3±27.9 |
| | Plot S-E | Open field | 24 ± 33 | 4.1±1.7 | 159±18 | 4.1 | 0.3±0.1 | 18.3±8.5 | 98.0±138.4 |
| Temperate forest | Plot T-A | Chinese pine forest | -0.04±0.81 | 2.22±0.87 | 72±12 | 5.8 | 17.0±8.55 | 9.77±6.57 | 17.09±29.4 |
| | Plot T-B | Larch forest | 0.32±0.96 | 2.30±0.94 | 141±15 | 25 | 26.3±6.51 | 10.0±6.23 | 22.9±18.6 |
| | Plot T-C | Wetland | 3.81±0.52 | 2.47±0.92 | 156±21 | 47 | 42.9±8.22 | 10.0±6.55 | 22.1±19.4 |
| | Plot T-D | Mixed broad-leaved forest | 0.68±1.01 | 2.37±0.87 | 74±9 | 16 | 25.4±7.32 | 9.86±6.26 | 25.9±18.6 |
| | Plot T-E | Open field | 1.82±0.79 | 1.98±0.79 | 52±4 | 12 | 27.9±5.56 | 10.1±6.47 | 47.1±29.4 |

821

822

**Figure captions:**

**Fig. 1.** Location of the five sampling plots and the estimation of soil-air fluxes (SA fluxes, values as g m$^{-2}$ yr$^{-1}$) at the temperate and subtropical forest. Potential vegetation of China is from the Vegetation Map of China (Hou, 1982). Up and down arrows represent emission and deposition, respectively.

**Fig. 2.** Mean and standard deviations of soil-air TGM fluxes at the five plots for the four seasons and annual values during the study at the subtropical forest (A) and temperate forest (B). The number of flux observations in spring, summer, autumn and winter were 62, 92, 66 and 43 at the subtropical forest and 60, 58, 60 and 14 for the temperate forest, respectively.

**Fig. 3.** Daily (average flux of day and night) composite Hg flux, solar radiation and soil temperature at Masson pine forests plots ((A) and (B)), wetland (C), evergreen broad-leaved forest (D) and open field (E) plots at the subtropical forest. The vertical arrows represent precipitation events.

**Fig. 4.** Daily (average flux of day and night) composite Hg flux, solar radiation and soil temperature at Chinese pine forest (A), larch forest (B), wetland (C), mixed broad-leaved forest (D) and open field (E) plots at the temperate forest. The vertical arrows represent precipitation events.

**Fig. 5.** Interplays of environmental factors on air-soil TGM exchange flux obtained by structural equation model (SEM) in the temperate (a) and subtropical (b) forests.

**Fig. 6.** Correlation between the air TGM concentration and air-surface Hg flux measured in daytime and night over four seasons for at Masson pine forest plots ((A) and (B)), wetland (C), evergreen broad-leaved forest (D) and open field (E) plots at the subtropical forest.

**Fig. 7.** Correlation between the air TGM concentration and air-surface Hg flux measured in daytime and night over four seasons for the five plots at Chinese pine forest (a), larch forest (b), wetland (c), mixed broad-leaved forest (d) and open field (e) plots at the temperate forest.

**Fig. 8.** Diurnal patterns of soil Hg fluxes with meteorological parameters in spring (a), summer (b), autumn (c) and winter (d) at the coniferous forest of the subtropical forest.

**Fig. 9.** Diurnal patterns of soil Hg fluxes with meteorological parameters in spring (a), summer (b), autumn (c) and winter (d) at the deciduous broad-leaved forest of the temperate forest.


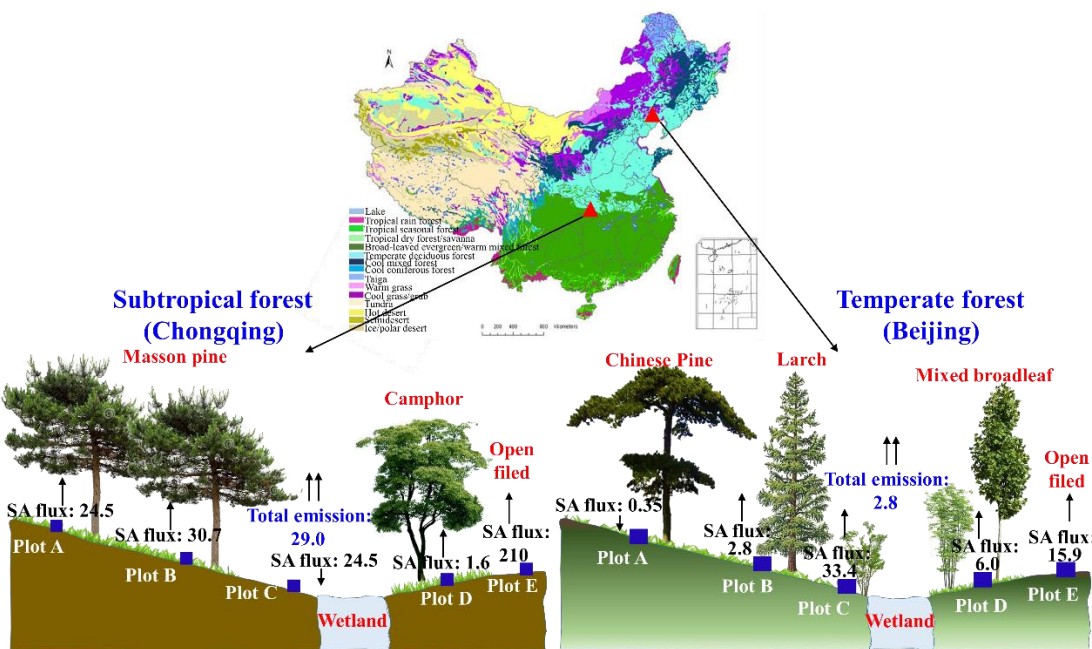


**Fig. 1.** Location of the five sampling plots and the estimation of soil-air fluxes (SA fluxes, values
as g m$^{-2}$ yr$^{-1}$) at the temperate and subtropical forest. Potential vegetation of China is from the
Vegetation Map of China (Hou, 1982). Up and down arrows represent emission and deposition,
respectively.


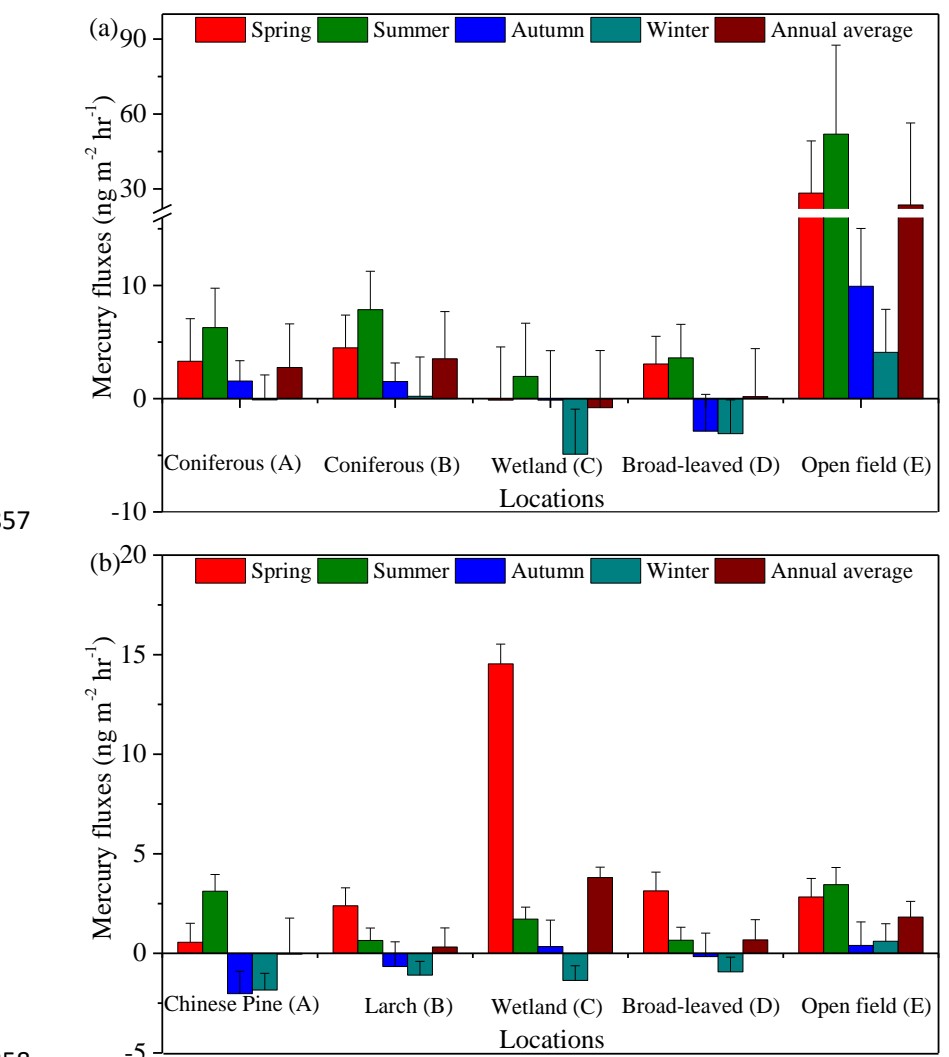


**Fig. 2.** Mean and standard deviations of soil-air TGM fluxes at the five plots for the four seasons and annual values during the study at the subtropical forest (A) and temperate forest (B). The number of flux observations in spring, summer, autumn and winter were 62, 92, 66 and 43 at the subtropical forest and 60, 58, 60 and 14 for the temperate forest, respectively.


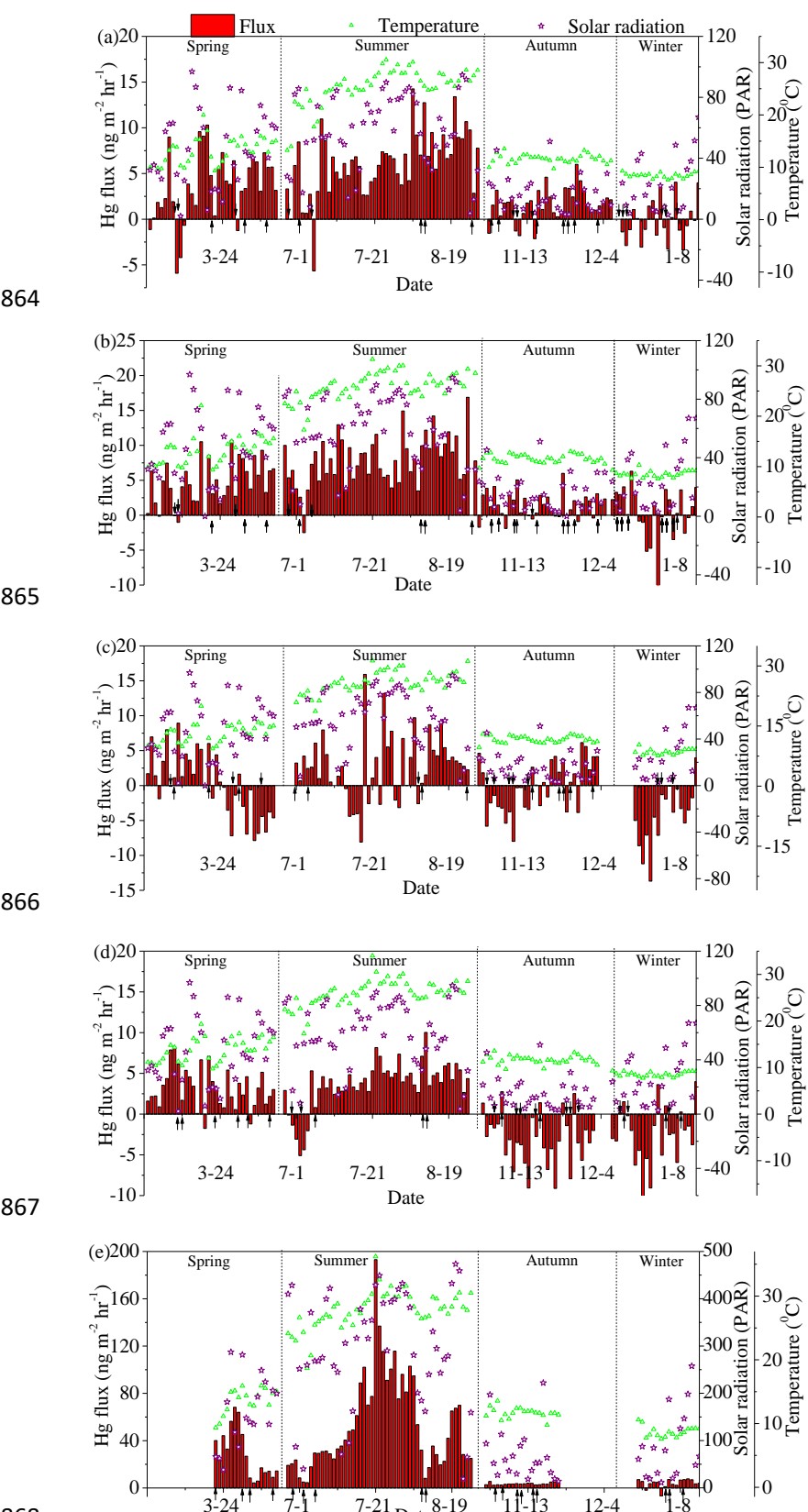






**Fig. 3.** Daily (average flux of day and night) composite Hg flux, solar radiation and soil temperature

at Masson pine forests plots ((A) and (B)), wetland (C), evergreen broad-leaved forest (D) and open

field (E) plots at the subtropical forest. The vertical arrows represent precipitation events.


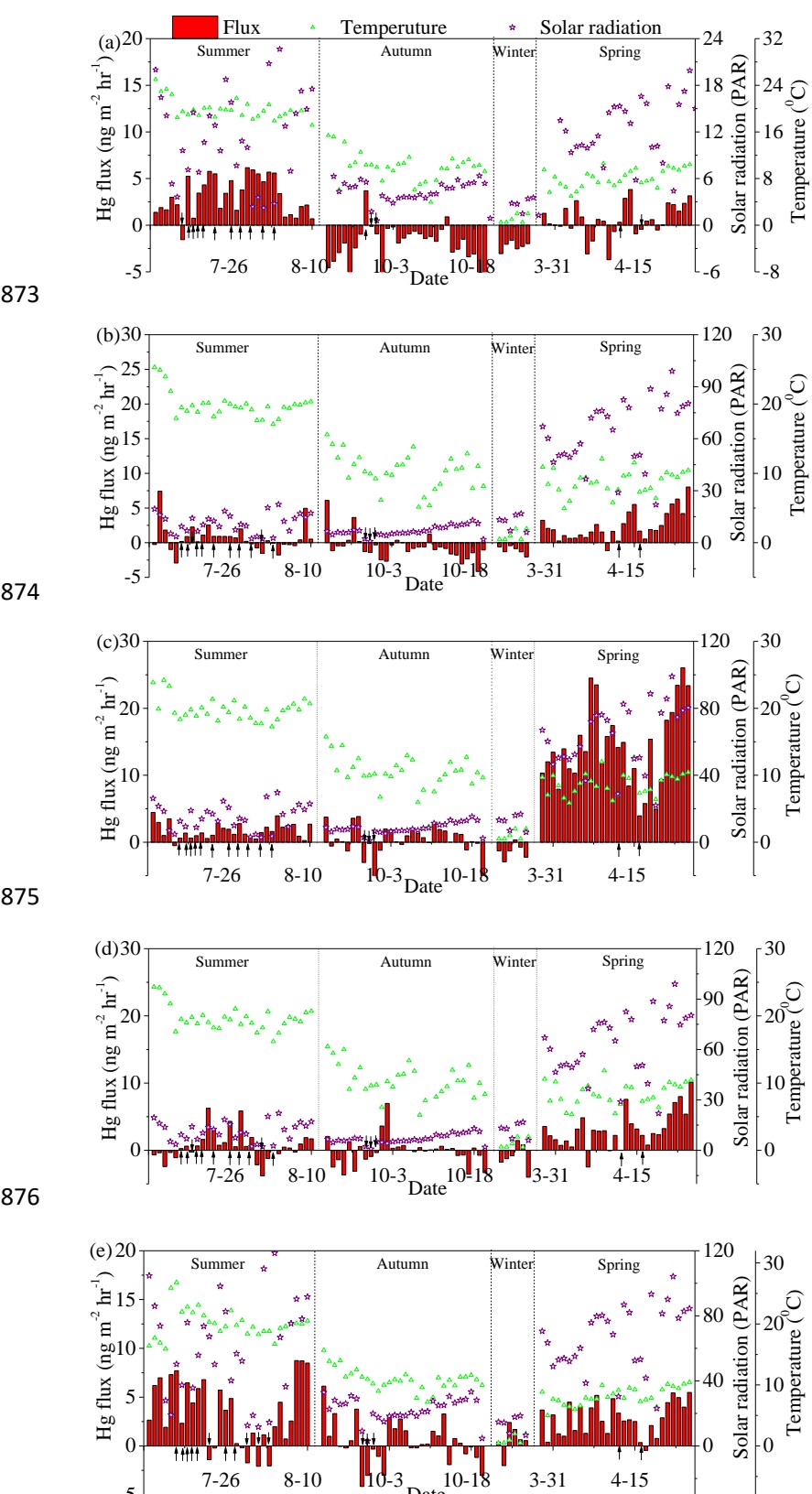

**Fig. 4.** Daily (average flux of day and night) composite Hg flux, solar radiation and soil temperature at Chinese pine forest (A), larch forest (B), wetland (C), mixed broad-leaved forest (D) and open field (E) plots at the temperate forest. The vertical arrows represent precipitation events.

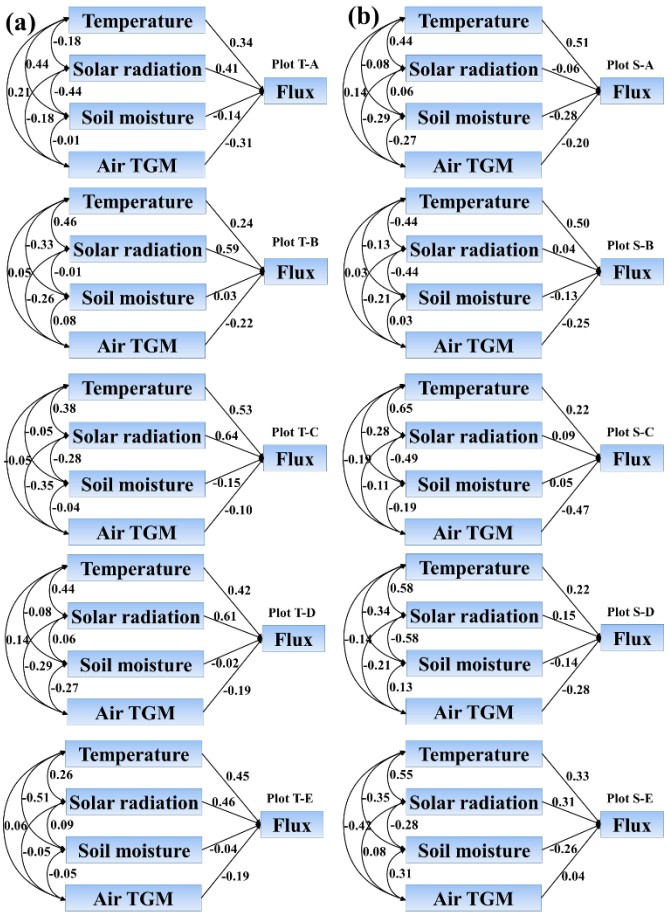


**Fig. 5.** Interplays of environmental factors on air-soil TGM exchange flux obtained by structural
equation model (SEM) in the temperate (a) and subtropical (b) forests.

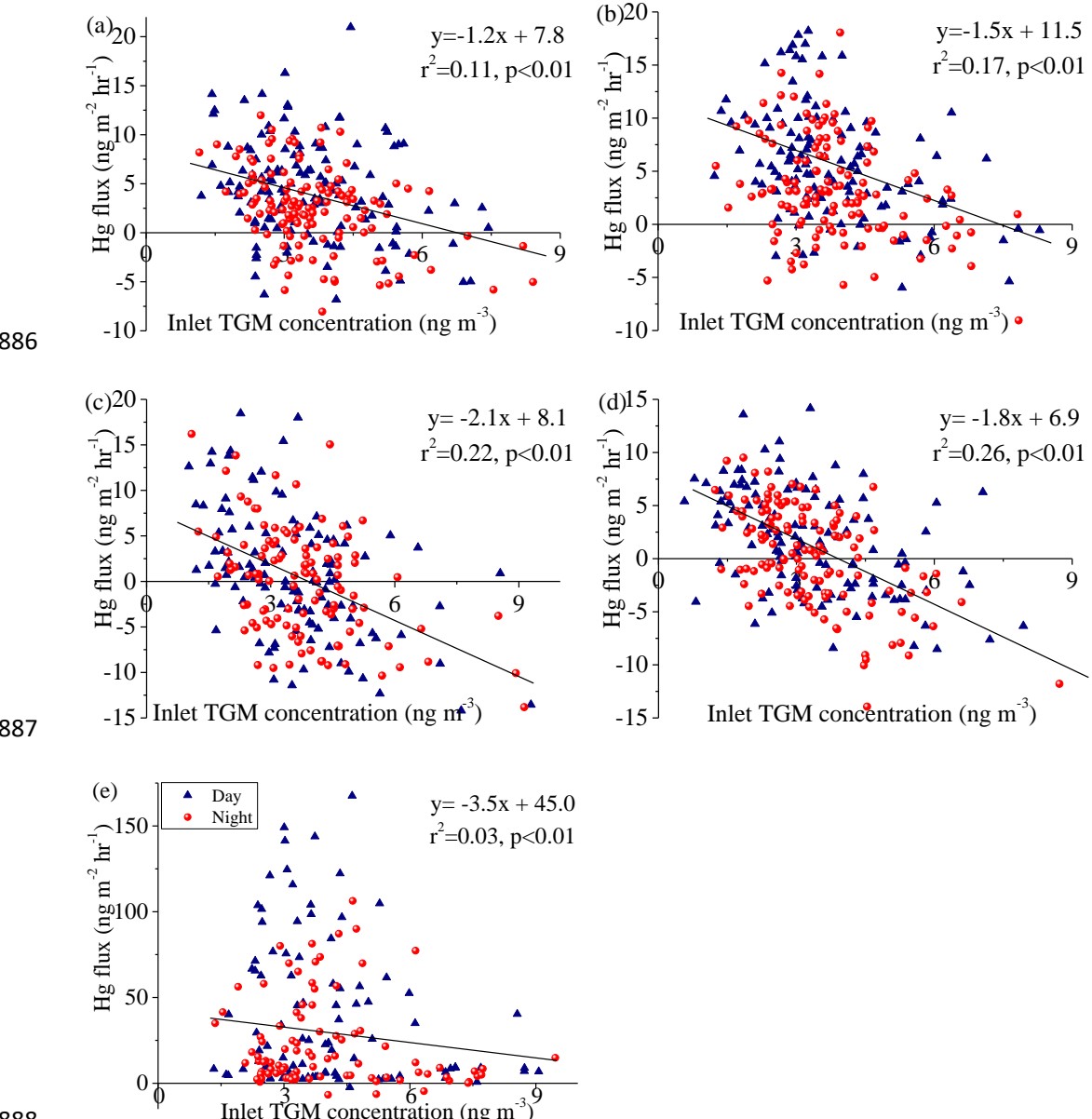




**Fig. 6.** Correlation between the air TGM concentration and air-surface Hg flux measured in daytime
and night over four seasons for at Masson pine forest plots ((A) and (B)), wetland (C), evergreen
broad-leaved forest (D) and open field (E) plots at the subtropical forest.


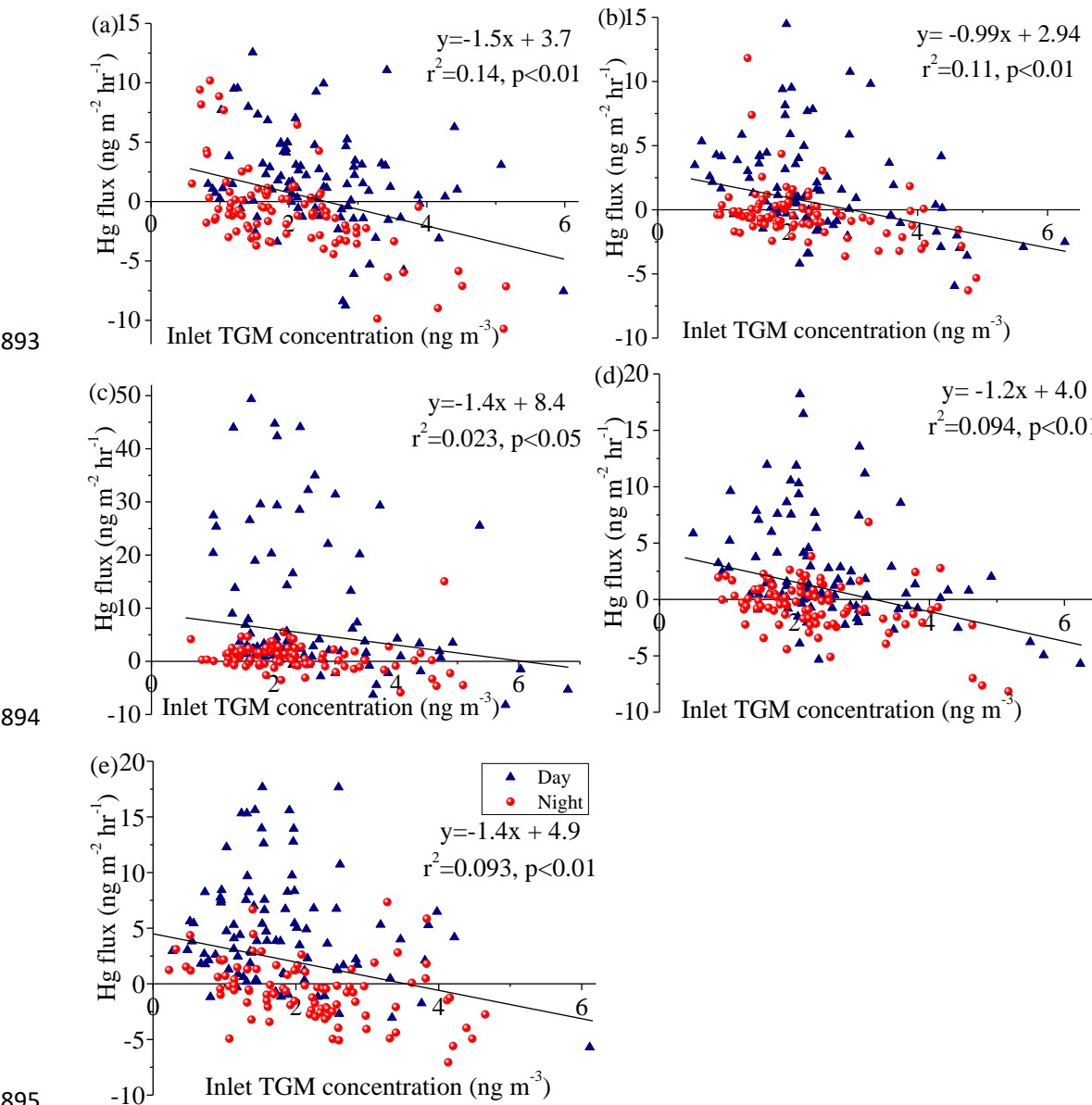




**Fig. 7.** Correlation between the air TGM concentration and air-surface Hg flux measured in daytime

and night over four seasons for the five plots at Chinese pine forest (a), larch forest (b), wetland (c),

mixed broad-leaved forest (d) and open field (e) plots at the temperate forest.


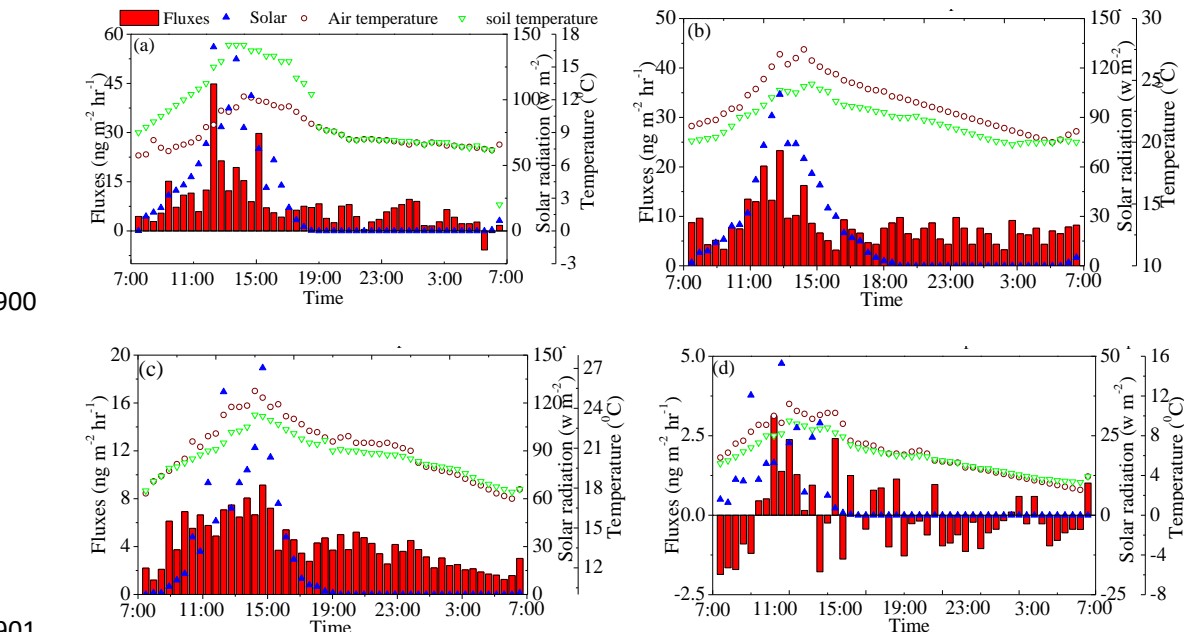



**Fig. 8.** Diurnal patterns of soil Hg fluxes with meteorological parameters in spring (a), summer (b),
autumn (c) and winter (d) at the coniferous forest of the subtropical forest.


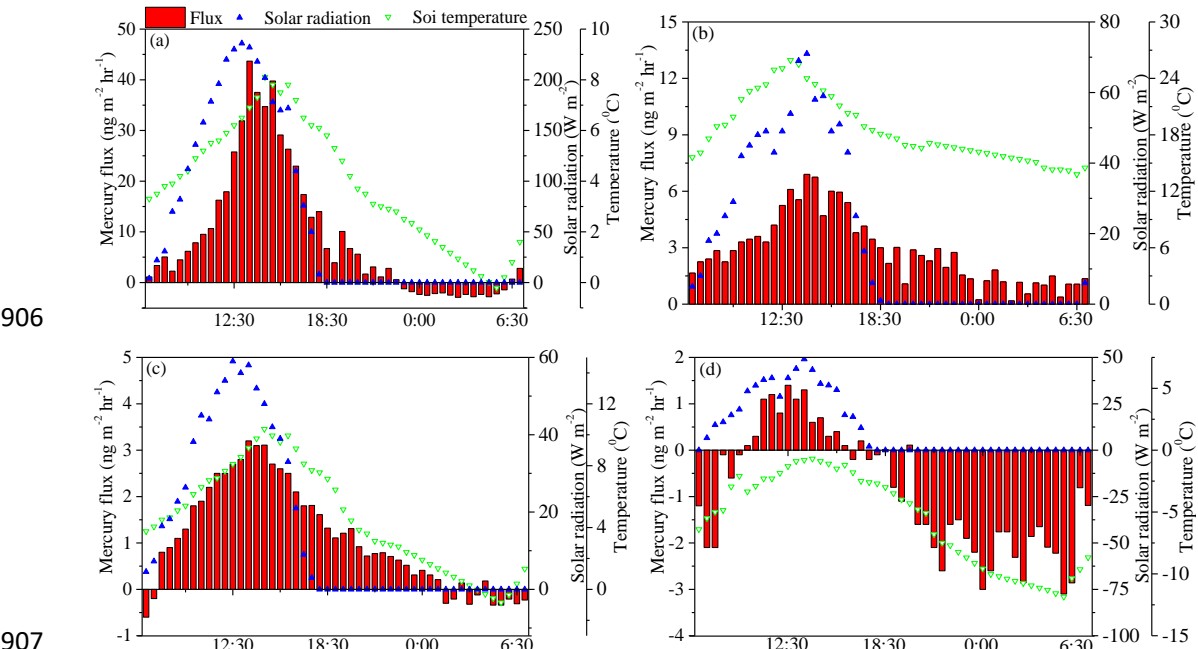

**Fig. 9.** Diurnal patterns of soil Hg fluxes with meteorological parameters in spring (a), summer (b), autumn (c) and winter (d) at the deciduous broad-leaved forest of the temperate forest.
