# Peer review of "and temperate forest catchments"

_Atmospheric Chemistry and Physics, 2020_

## Referee Comment (RC1) · Anonymous Referee #1 · 24 Sep 2020

Zhou et al., studied Hg evasion from a subtropical forest and a temperate forest, and they found that fluxes showed strong positive relationships with solar radiation and soil temperature, and negative correlations with ambient-air TGM concentration in both subtropical and temperate forests. They highlighted more attention should pay to the legacy Hg stored in terrestrial surface as a more important increasing Hg emission source with the decreasing air TGM concentration recently. Generally, this study demonstrates some interesting observation in forest air-soil flux exchanges, and these new finding can help us to better understand the Hg fluxes. But I have some concerned issues need the authors to further polish this manuscript before accept. (1) Many studies have suggested that solar radiation and soil temperature have strong ef-

fects to induce soil Hg evasion from soil. Authors also have stated these earlier studies results. To me, I am not surprised these results. There are significant correlations among temperature and solar radiation. The effects of these environmental factors on Hg0 flux are confounded. However, the synergistic effect from multiple factors leads to hard determine the individual effect of each parameter. Recently, I also read a sub-tropical forest air-soil Hg0 flux study in China (Yuan, Wei; Wang, Xun; Lin, Che-Jen.; Sommar, Jonas; Lu, Zhiyun; Feng, Xinbin, Process factors driving dynamic exchange of elemental mercury vapor over soil in broadleaf forest ecosystems. Atmos Environ 2019, 219, 117047). They used SEM equations to demonstrate the temperature is the key parameter to shape the soil Hg0 evasion. I wonder does temperature play the sim-ilar role in this study as Yuan's study, and I also suggest using similar SEM to further demonstrate the effects from atmospheric Hg0, landuse, environmental parameters. (2) There are several forest air-soil Hg fluxes studies in subtropical regions in China, such as Yuan 2019, and Yu et al., 2020 (Subtropical Forests Act as Mercury Sinks but as Net Sources of Gaseous Elemental Mercury in South China, Environ Sci Technol). I suggest authors should compare their results to those studies to support your several hypotheses. (3) The most interesting results in this study is that air-soil flux varies with the landuse, and distinctly different compensation point for each landuse. However, authors just depicted these results without further explanation and hypothesis.

Line 24, "estimates" grammar wrong. Line 25, "soil-atmosphere exchange, soil-air gaseous Hg" why repeat twice? Line 27-28, "showed patterns of both emission and deposition at five study plots, with an area-weighted net emission rate of 3.2 and 0.32 ng m−2 hr−1 for the entire subtropical and temperate forests, respec-tively". This sentence is confused, which forest is a Hg sink or source? Line 29-31 rephase this sentence because of very hard to understand. Line 35 rephase "at" to "in" Line 51 rephase this sentence because of unclear Line 94 I did get your logic flow here when authors stated " serve as sources of previously deposited Hg". Line 101, I recently read several subtropical forest studies in China, and authors stated "scarce" is not right. Line 116. Wrong sentence for "Dongling (MDL). . ."

---

## Referee Comment (RC2) · Anonymous Referee #2 · 27 Sep 2020

The authors report flux measurements of total gaseous mercury (TGM) on 5 plots in subtropical forest and 5 plots in temperate forest in four seasons. They use the dynamic flow chamber (DFC) method and describe the flux dependence on ambient TGM concentrations, solar radiation, and temperature. The diurnal variations in different seasons are described.

The measurements are valuable but the authors stretch their interpretation by taking the measured fluxes as being representative for the whole investigated ecosystems. DFC measurements are well suited to study the flux mechanism, i.e. flux dependence on temperature, soil moisture, ambient TGM concentration, solar radiation, soil tem-

perature, substrate concentrations, etc. But they are unsuitable for determination of the representative fluxes for a given ecosystem because a) only a small area is being measured (20 x 30 cm here) and b) covering of the soil by DFC changes its status (e.g. by heating the soil or vegetation by glasshouse effect). In other words: really representative fluxes have to be measured by micrometeorological methods, DFC methods can provide only empirical relationships for extrapolating them to the whole ecosystems. The problem with this paper is that the authors try to estimate ecosystem fluxes as if their measurements were representative for them, despite being aware of the problems in DFC measurements (mentioned in meagre 3-4 lines).

I recommend the publication of the paper provided that the authors stick with the mechanistical interpretation of their results and avoid the temptation of extrapolations to the whole ecosystems (made e.g. in "Conclusions and study implications)". This would need some changes in the text. The authors also discuss the observed correlations and relations predominantly in physicochemical terms. By this they neglect the soil microbiology – this also needs to be rectified.

Specific comments:

Line 50: "..long longevity... is able to undergo over long distances.."?

Line 59: Not all fires are "natural".

Lines 147-149: "semi-cylindical" and "20 x 30 cm" – how does it fit together? "Six inlet holes" where?

Lines 147-153: How was the chamber installed on the soil: was it partly buried into the soil to seal the chamber-soil gap, if so to which depth? Are you sure that you do not suck ambient air through the soil or through the gap between the chamber and the soil, at least partly, instead of sucking air through the inlet holes? The resistance of the soil with respect to air flow can be surprisingly small, it may be smaller than the resistance of the inlet holes, resulting in sucking of air through the soil. If that happens,

the measured fluxes are not what was intended to be measured. Eckley et al. (2010) do not mention this problem.

If I understand the text properly then the chambers were permanently (during the measurement period) on the soil. If so, then the plot under measurement would e.g. not receive any precipitation? In other words: the measurements would not be representative for uncovered soil. Please specify.

Gold cartridges: what type? Those of Tekran or other? Please specify.

Lines 176-178: In these few lines the authors mention the problems with fluxes measured by DFC and, essentially, salvage themselves using Eckley et al. (2010) reference. The chapter "Conclusions and study implications" is written as if there were no problems.

Line 185: soil organic matter (SOM)

Line 194: Sampling TGM in pore air is mentioned – how was it made? What were the results?

Line 267: were

Lines 273-279: The influence of soil humidity is discussed here only in terms of physicochemical terms. It is well known that microbiological processes in dry soils are greatly enhanced by occasional precipitation.

Line 296: "physicochemical properties" – what about microbiological ones?

Line 368: Photo-reduction of $Hg^{2+}$ may be a major driver in waters but hardly in soils which are impenetrable to solar radiation. More plausible is the explanation by higher soil temperature and the related higher microbiological activity.

Paragraph, lines 417-433, Figure 5: Are these correlations made with data from all seasons? I would expect different compensation points for different seasons.

[Figure]

Figures 3 and 4: What does the x axis mean?

Figures 5 and 6: Are these plots seasonally resolved? If not please state that data from all seasons were used.

SI, "Environmental measurements": The measurements of soil temperature (depth) is not mentioned here and neither in "Experimental". This parameter is the crucial one for physicochemical and microbiological processes in the soil. According to Figure S 6 it seems to have been measured. I would prefer to discuss all relationships in relation to soil temperature instead of solar radiation. Solar radiation is essentially only a sort of proxy parameter for soil temperature. It is also not applicable for the night.

SI, description of MDL: Any information about the Hg content of litterfall and soil?

---

## Author Comment (AC1) · 11 Nov 2020

**Reply to Comments from Reviewer #1**

We appreciate the constructive and thoughtful comments from the reviewers which have helped us improve the manuscript. We have carefully revised our manuscript following the reviewer's comments. A point-to-point response is given below. The reviewers' comments are in black and our replies are in blue.

**To reviewer**

*Comment 1:*

Zhou et al., studied Hg evasion from a subtropical forest and a temperate forest, and they found that fluxes showed strong positive relationships with solar radiation and soil temperature, and negative correlations with ambient-air TGM concentration in both subtropical and temperate forests.

They highlighted more attention should pay to the legacy Hg stored in terrestrial surface as a more important increasing Hg emission source with the decreasing air TGM concentration recently.

Generally, this study demonstrates some interesting observation in forest air-soil flux exchanges, and these new finding can help us to better understand the Hg fluxes. But I have some concerned issues need the authors to further polish this manuscript before accept.

Response: We thank the reviewer's constructive comments on this manuscript. We have addressed all the reviewer's concerned issues below. We hope the revised manuscript can meet the standards for publication in *Atmospheric Chemistry and Physics*.

*Comment 2:*

Many studies have suggested that solar radiation and soil temperature have strong effects to induce soil Hg evasion from soil. Authors also have stated these earlier studies results. To me, I am not surprised these results. There are significant correlations among temperature and solar radiation.

The effects of these environmental factors on Hg0 flux are confounded. However, the synergistic effect from multiple factors leads to hard determine the individual effect of each parameter. Recently,

I also read a subtropical forest air-soil Hg0 flux study in China (Yuan, Wei; Wang, Xun; Lin, Che-

Jen.; Sommar, Jonas; Lu, Zhiyun; Feng, Xinbin, Process factors driving dynamic exchange of elemental mercury vapor over soil in broadleaf forest ecosystems. Atmos Environ 2019, 219,

117047). They used SEM equations to demonstrate the temperature is the key parameter to shape the soil Hg0 evasion. I wonder does temperature play the similar role in this study as Yuan's study, and I also suggest using similar SEM to further demonstrate the effects from atmospheric Hg0, land use, environmental parameters.

Response: We appreciate the reviewer's suggestions. We agree that the synergistic effects from multiple factors makes in difficult evaluate the effect of each factor. The SEM approach was applied the observations in this study following the reviewer's suggestion The results of SEM is shown in

Fig. 5 and is described in the revised manuscript on lines 239-242, 411-416, 461-463, and 473-475:

[Figure]

**Fig. 5.** The interplay of environmental factors on air-soil TGM exchange fluxes determined by structural equation model (SEM) in the temperate (a) and subtropical (b) forests.

"Structural equation modeling (SEM) were performed on the collected Hg flux data using Amos software. SEM, developed from a fully conceptual model using $\chi$2 tests with maximum likelihood estimation, was conducted to infer the interplay of temperature, solar radiation, soil moisture, and air TGM concentrations on measurements of soil-air TGM exchange fluxes."

"To consider synergistic effects from multiple factors, SEM was applied to infer the soil-air TGM exchange processes (Fig. 5). It is clear that temperature was a more dominant factor driving air-soil TGM exchange flux over the four seasons in the subtropical forest plots, while solar radiation was a more dominant factor at the temperate forest due to direct exposure of the forest floor to solar radiation the leaf-off seasons. At the open fields of both forests, temperature and solar radiation had a synergistic effect on soil Hg fluxes."

"Soil-air Hg fluxes also showed significant negative correlations with atmospheric TGM concentrations at the ten plots at both forests ($r^2 = 0.023-0.26$, $p < 0.05$, Fig. 6 and 7), which had a greater effect than soil moisture at both forests, except for plots T-C, S-A and S-E (Fig. 5)."

"SEM inferred that that air TGM concentrations was the second important driver influencing the soil-air TGM exchange in Masson pine (Plot S-B), evergreen broad-leaved and wetland plots at subtropical forest (Fig. 5)."

***Comment 3:***

There are several forest air-soil Hg fluxes studies in subtropical regions in China, such as Yuan 2019, and Yu et al., 2020 (Subtropical Forests Act as Mercury Sinks but as Net Sources of Gaseous Elemental Mercury in South China, Environ Sci Technol). I suggest authors should compare their results to those studies to support your several hypotheses.

Response: We thank the reviewer for these suggestions. We have added the text comparing our results with Yuan et al. 2019, Yu et al. 2020 and some other studies, these modification are added in the lines 105-113 and 416-420:

"Forest ecosystems not only act as sinks for atmospheric Hg deposition, but can also serve as sources resulting from legacy Hg that has accumulated in surface soil. For example, one study constructed the Hg budget in subtropical forest in southern China showing that the forest is a minor sink for atmospheric Hg but a significant net Hg(0) source (58.5 μg m$^{-2}$ yr$^{-1}$) (Yu et al., 2020). In contrast, another study also in southern China using budgets of air-foliage and air-soil Hg(0) exchange fluxes, showed that forest is a net sink of Hg(0) (20.1 μg m$^{-2}$ yr$^{-1}$) (Yuan et al., 2019a;Yuan et al., 2019b). These results indicate that there is considerable uncertainty and variability in the source-sink behavior of Hg in subtropical forests of southern China. Furthermore, no studies have conducted in northern China to characterize the Hg fluxes in the temperate forest."

"A recent study of soil-air TGM fluxes at subtropical evergreen broadleaf forest in South China also suggested that temperature is the most important driver of air-soil TGM exchange (Yuan et al., 2019b). Therefore, we may infer that under the shade of the forest canopy, temperature is the dominant factor causing variation in TGM evasion from forest soil."

*Comment 4:*

The most interesting results in this study is that air-soil flux varies with the landuse, and distinctly different compensation point for each landuse. However, authors just depicted these results without further explanation and hypothesis.

Response: We have added discussion on the implications of changes in the forest stand landscape and climate on soil mercury dynamics on lines 516-528:

"A recent study using models simulating the dynamics of the subtropical forest landscape under climate change, harvesting, and land-use disturbances in southern China showed that coniferous forest area increased approximately 3.7 times compared to broad-leaved forest area (Wu et al., 2019). In the temperate forest, climatic changes in the northern China are expected to cause coniferous stands to transition to deciduous forests over the next hundred years (Ma et al., 2014). Climate change and land-use disturbance may increase the compensation points in both temperate and subtropical forests, therefore, increasing emissions of legacy Hg from terrestrial soils to the atmosphere. Some studies have emphasized that climate and land use change will potentially enhance deposition of Hg to forested landscapes (Haynes et al., 2017;Richardson and Friedland, 2015;Li et al., 2020); however, our study suggests that legacy Hg in forest soils could be emitted back to atmosphere, offsetting enhanced atmospheric Hg deposition. Better understanding of the response of Hg emissions from forest soils to climate and land use change is an important topic for future research."

*Comment 5:*

Line 24, "estimates" grammar wrong.

Response: The word has been revised to "estimate"

*Comment 6:*

105  Line 25, "soil-atmosphere exchange, soil-air gaseous Hg" why repeat twice?

106 Response: The "soil-atmosphere exchange" has been deleted.

108 ***Comment 7:***

109  Line 27-28, "showed patterns of both emission and deposition at five study plots, with an area-

110 weighted net emission rate of 3.2 and 0.32âAL'ngâ ˘ AL'm ˘ -2âAL'hr ˘ -1 for the entire subtropical

111 and temperate forests, respectively". This sentence is confused, which forest is a Hg sink or source?

112 Response: We have deleted the sentence and added values to describe the deposition or emission

113 values of the plots in line 28-36:

114  "At the subtropical forest the highest net soil Hg emissions were observed for an open field (24

115 $\pm$ 33 ng m$^{-2}$ hr$^{-1}$), followed by two coniferous forest plots (2.8 $\pm$ 3.9 and 3.5 $\pm$ 4.2 ng m$^{-2}$ hr$^{-1}$),

116 broad-leaved forest plot (0.18 $\pm$ 4.3 ng m$^{-2}$ hr$^{-1}$), and the remaining wetland site showing net

117 deposition ($-0.80 \pm 5.1$ ng m$^{-2}$ hr$^{-1}$). At the temperate forest, the highest fluxes and net soil Hg

118 emissions were observed for a wetland (3.81 $\pm$ 0.52 ng m$^{-2}$ hr$^{-1}$) and an open field (1.82 $\pm$ 0.79 ng

119 m$^{-2}$ hr$^{-1}$), with lesser emission rates in deciduous broad-leaved forest (0.68 $\pm$ 1.01 ng m$^{-2}$ hr$^{-1}$) and

120 deciduous needle-leaved forest (0.32 $\pm$ 0.96 ng m$^{-2}$ hr$^{-1}$) plots, and net deposition at an evergreen

121 pine forest ($-0.04 \pm 0.81$ ng m$^{-2}$ h$^{-1}$)."

123 ***Comment 8:***

124  Line 29-31 rephrase this sentence because of very hard to understand.

125 Response: The sentence has been rephrased in line 28-32:

126  "At the subtropical forest the highest net soil Hg emissions were observed for an open field (24

127 $\pm$ 33 ng m$^{-2}$ hr$^{-1}$), followed by two coniferous forest plots (2.8 $\pm$ 3.9 and 3.5 $\pm$ 4.2 ng m$^{-2}$ hr$^{-1}$),

128 broad-leaved forest plot (0.18 $\pm$ 4.3 ng m$^{-2}$ hr$^{-1}$), and the remaining wetland site showing net

129 deposition ($-0.80 \pm 5.1$ ng m$^{-2}$ hr$^{-1}$)."

131 ***Comment 9:***

132 Line 35 rephrase "at" to "in"

133 Response: The text was changed accordingly in line 37.

***Comment 10:***

Line 51 rephrase this sentence because of unclear

Response: The sentence has been rephrased in line 54-56.

"Hg(0) is relatively inert and has a long atmospheric lifetime of 0.5–1 year, which allows for long range transport (Kamp et al., 2018;Slemr et al., 2018;St Louis et al., 2019)."

***Comment 11:***

Line 94 I did get your logic flow here when authors stated "serve as sources of previously deposited Hg".

Response: The sentence has been rephrased in line 105-106:

"Forest ecosystems not only act as sinks for atmospheric Hg deposition, but can also serve as sources resulting from legacy Hg that has accumulated in surface soil."

***Comment 12:***

Line 101, I recently read several subtropical forest studies in China, and authors stated "scarce" is not right.

Response: We have deleted the sentence.

***Comment 13:***

Line 116. Wrong sentence for "Dongling (MDL)..."

Response: The sentence has been rephrased in line 129-130:

[revised manuscript text omitted]

under the forest canopy at the subtropical forest. Standard deviations of soil Hg concentrations were obtained from Hg concentrations over the four seasons (n=12). Because fluxes are often controlled by solar radiation for bare soils, the correlation analysis above does not include data from the open field (plot E).

**Fig. S6.** Soil-air TGM fluxes during the daytime and nighttime at Masson pine forests (Plot A) and (Plot B), wetland (Plot C), evergreen broad-leaved forest (Plot D) and open field (Plot E) at the subtropical forest (a), and at Chinese pine forest (Plot A), larch forest (Plot B), wetland (Plot C), mixed broad-leaved forest (Plot D) and open field (Plot E) at the temperate forest (b).

**Fig. S7.** Correlations between soil temperature and air-surface Hg fluxes measured during daytime and night at the Masson pine forests (a) and (b), wetland (c), evergreen broad-leaved forest (d)

and open field (e) in the subtropical forest.

**Fig. S8.** Correlations between soil temperature and air-surface Hg fluxes measured during daytime and night at the Chinese pine forest (a), larch forest (b), wetland (c), mixed broad-leaved forest (d) and open field (e) at the temperate forest.

**Fig. S9.** Correlations between soil moisture and air-surface Hg fluxes measured during daytime and night at the Chinese pine forest (a), larch forest (b), wetland (c), mixed broad-leaved forest (d)

and open field (e) at the subtropical forest.

**Fig. S10.** Correlations between soil moisture and air-surface Hg fluxes measured during daytime and night at the Chinese pine forest (a), larch forest (b), wetland (c), mixed broad-leaved forest (d) and open field (e) at the temperate forest.

**Fig. S11.** Correlations between the gradient of Hg(0) concentrations between surface soil pore (at 3

cm) and atmospheric values and soil-air Hg(0) flux at four plots at the subtropical forest.

**Fig. S12.** Correlations between the gradient of Hg(0) concentrations between surface soil pore (at 3

cm) and atmospheric values and soil-air Hg(0) flux at the four plots at the temperate forest.

[Figure]

**Fig. S1.** Schematic diagram of the dynamic flux chamber used in this study.

[Figure]

**Fig. S2.** Correlations between the averaged solar radiation (8:00-17:00) and air-surface Hg flux measured during daytime in Masson pine forests (a) and (b), wetland (c), evergreen broad-leaved forest (d) and open field (e) in the subtropical forest.

[Figure]

**Fig. S3.** Correlation between the averaged solar radiation (8:00-17:00) and air-surface Hg flux
measured during daytime in Chinese pine forest (a), larch forest (b), wetland (c), mixed broad-
leaved forest (d) and open field (e) in the temperate forest.

[Figure]

**Fig. S4.** Effects of rainfall events on annual soil-air TGM fluxes at Masson pine forests (Plot A) and (Plot B), wetland (Plot C), evergreen broad-leaved forest (Plot D) and open field (Plot E) at the subtropical forest (A), and at Chinese pine forest (Plot A), larch forest (Plot B), wetland (Plot C), mixed broad-leaved forest (Plot D) and open field (Plot E) at the temperate forest (B).

[Figure]

**Fig. S5.** Correlation between the soil Hg concentrations ($S_c \pm SD$) and soil-air Hg flux ($F \pm SD$) under the forest canopy at the subtropical forest. Standard deviations of soil Hg concentrations were obtained from Hg concentrations over the four seasons (n=12). Because fluxes are often controlled by solar radiation for bare soils, the correlation analysis above does not include data from the open field (plot E).

[Figure]

**Fig. S6.** Soil-air TGM fluxes during the daytime and nighttime at Masson pine forests (Plot A) and
(Plot B), wetland (Plot C), evergreen broad-leaved forest (Plot D) and open field (Plot E) at the
subtropical forest (a), and at Chinese pine forest (Plot A), larch forest (Plot B), wetland (Plot C),
mixed broad-leaved forest (Plot D) and open field (Plot E) at the temperate forest (b).

[Figure]

**Fig. S7.** Correlations between soil temperature and air-surface Hg fluxes measured during daytime and night at the Masson pine forests (a) and (b), wetland (c), evergreen broad-leaved forest (d) and open field (e) in the subtropical forest.

[Figure]

s

**Fig. S8.** Correlations between soil temperature and air-surface Hg fluxes measured during daytime
and night at the Chinese pine forest (a), larch forest (b), wetland (c), mixed broad-leaved forest (d)
and open field (e) at the temperate forest.

[Figure]

**Fig. S9.** Correlations between soil moisture and air-surface Hg fluxes measured during daytime and night at the Chinese pine forest (a), larch forest (b), wetland (c), mixed broad-leaved forest (d) and open field (e) at the subtropical forest.

[Figure]

**Fig. S10.** Correlations between soil moisture and air-surface Hg fluxes measured during daytime
and night at the Chinese pine forest (a), larch forest (b), wetland (c), mixed broad-leaved forest (d)
and open field (e) at the temperate forest.

[Figure]

**Fig. S11.** Correlations between the gradient of Hg(0) concentrations between surface soil pore (at 3

cm) and atmospheric values and soil-air Hg(0) flux at four plots at the subtropical forest.

[Figure]

**Fig. S12.** Correlations between the gradient of Hg(0) concentrations between surface soil pore (at 3 cm) and atmospheric values and soil-air Hg(0) flux at the four plots at the temperate forest.

---

## Author Comment (AC2) · 11 Nov 2020

**Reply to Comments from Reviewer #2**

We appreciate the constructive and thoughtful comments from the reviewers which have helped us improve the manuscript. We have carefully revised our manuscript following the reviewer's comments. A point-to-point response is given below. The reviewers' comments are in black and our replies are in blue.

**To reviewer**

*Comment 1:*

The authors report flux measurements of total gaseous mercury (TGM) on 5 plots in subtropical forest and 5 plots in temperate forest in four seasons. They use the dynamic flow chamber (DFC)

method and describe the flux dependence on ambient TGM concentrations, solar radiation, and temperature. The diurnal variations in different seasons are described.

The measurements are valuable but the authors stretch their interpretation by taking the measured fluxes as being representative for the whole investigated ecosystems. DFC measurements are well suited to study the flux mechanism, i.e. flux dependence on temperature, soil moisture, ambient TGM concentration, solar radiation, soil temperature, substrate concentrations, etc. But they are unsuitable for determination of the representative fluxes for a given ecosystem because a)

only a small area is being measured (20 x 30 cm here) and b) covering of the soil by DFC changes its status (e.g. by heating the soil or vegetation by glasshouse effect). In other words: really representative fluxes have to be measured by micrometeorological methods, DFC methods can provide only empirical relationships for extrapolating them to the whole ecosystems. The problem with this paper is that the authors try to estimate ecosystem fluxes as if their measurements were representative for them, despite being aware of the problems in DFC measurements (mentioned in meagre 3-4 lines).

I recommend the publication of the paper provided that the authors stick with the mechanistical interpretation of their results and avoid the temptation of extrapolations to the whole ecosystems (made e.g. in "Conclusions and study implications)". This would need some changes in the text.

The authors also discuss the observed correlations and relations predominantly in physicochemical terms. By this they neglect the soil microbiology – this also needs to be rectified.

Response: We thank the reviewer for providing constructive and thoughtful comments on our manuscript. We agree with the concern that there are some limitations of the use of DFCs to estimate ecosystem fluxes of Hg. We have deleted statements about estimates of whole ecosystem Hg fluxes in the revised paper, especially in the sections in the Abstract and Conclusions and study implications. This version of the paper now focuses on the mechanism of soil-air Hg exchange fluxes under different land cover conditions and discusses effects of temperature, incident solar radiation and precipitation on soil Hg exchange and implications of climate change induced transition of forest stands on Hg emissions. Additionally, we have also added discussion about the role of soil microbial transformations on TGM emissions as detailed in the comments below.

***Comment 2:***

Line 50: "..long longevity*: : :* is able to undergo over long distances.."?

Response: The sentence has revised in 54-56:

"Hg(0) is relatively inert and has a long atmospheric lifetime of 0.5–1 year, which allows for long range transport (Kamp et al., 2018;Slemr et al., 2018;St Louis et al., 2019).."

***Comment 3:***

Line 59: Not all fires are "natural".

Response: The text has been changed to "Although many studies have focused on primary anthropogenic Hg emissions, releases from natural source materials is also an important pathway but with greater uncertainty and variability, including emissions from natural reservoirs (e.g. volcanic activity, geothermal sources, weathering of Hg from soil minerals) and re-emissions of previous deposited Hg." in line 63-66.

***Comment 4:***

Lines 147-149: "semi-cylindical" and "20 x 30 cm" – how does it fit together? "Six inlet holes" where?

Response: We have added the schematic drawing of the flux chamber in the Fig. S1.

[Figure]

**Fig. S1.** Schematic diagram of the dynamic flux chamber used in this study.

*Comment 5:*

Lines 147-153: How was the chamber installed on the soil: was it partly buried into the soil to seal the chamber-soil gap, if so to which depth? Are you sure that you do not suck ambient air through the soil or through the gap between the chamber and the soil, at least partly, instead of sucking air through the inlet holes? The resistance of the soil with respect to air flow can be surprisingly small, it may be smaller than the resistance of the inlet holes, resulting in sucking of air through the soil. If that happens, the measured fluxes are not what was intended to be measured.

Eckley et al. (2010) do not mention this problem. If I understand the text properly then the chambers were permanently (during the measurement period) on the soil. If so, then the plot under measurement would e.g. not receive any precipitation? In other words: the measurements would not be representative for uncovered soil. Please specify.

Response: The chamber was not buried into the soil and placed on the top of the forest floor. To seal the chamber-soil gap, local fine soil was placed outside of the chamber bottom. We believe that the resistance of the fine soil (forest soil is relatively moisture) would be much higher than that of six inlets (1 cm in diameter) on the chambers. The chamber was not positioned in a fixed (during the measurement period) on the soil. Rather we moved the chambers to a new position as least every week when there was no rain and moved it to new position after days in which it rained. Therefore, the flux measurements are better able represent conditions under different weather. We have revised the text to clarify these approaches in lines 162-163 and 167-190:

"Local fine grained soil was placed outside the chamber to seal any gap between the base of the chamber and the soil."

"The DFC chambers in all plots were moved every week to mitigate against changes in soil moisture due the covering of soil by the chambers. If a precipitation event occurred, the chambers were also moved to new positions during the sampling period (morning or evening) to be representative of soil conditions receiving ambient precipitation."

***Comment 6:***

Gold cartridges: what type? Those of Tekran or other? Please specify.

Response: We made the gold cartridges in the laboratory and this is described in the revised text on line 166-169:

"All the gold cartridges were constructed with gold silk (< 0.5 mm diameter). The strands of gold silk were rolled together in a small coil and about 15 coils were used to fill a quartz cartridge with about 2 g of gold. The accuracy of all traps were evaluated (see section 2.4) and non- conforming cartridges were discarded."

***Comment 7:***

Lines 176-178: In these few lines the authors mention the problems with fluxes measured by

DFC and, essentially, salvage themselves using Eckley et al. (2010) reference. The chapter

"Conclusions and study implications" is written as if there were no problems.

Response: Following the reviewer's suggestion, we have deleted the text about the estimate of whole ecosystem Hg flux in the last section.

***Comment 8:***

Line 185: soil organic matter (SOM)

Response: The text has been changed accordingly on line 206.

***Comment 9:***

Line 194: Sampling TGM in pore air is mentioned – how was it made? What were the results?

Response: We sampled the TGM in the soil pore gas as part of this study, but not the results are on presented in this paper.    Initially we wanted to combine the DFC measurements with the soil pore

Hg results in a single comprehensive paper, but the reviewers felt this was too much material for a single paper. So we have split the results into two papers: this paper with the DFC results and a companion paper summarizing the soil Hg gas patterns. The companion manuscript on TGM

concentrations in the pore gas has been submitted to another journal. However we would like to make linkages between the two data sets and therefore have brought the soil pore Hg data into the discussion in this manuscript in Fig. S10 and S11 and line 467-470:

"In a companion study, the soil pore TGM concentrations were measured at all the plots at the subtropical and temperate forests, except the wetlands (Zhou et al., in review). These results showed that gradient of TGM concentrations between the surface air and pore air at 3 cm were significantly correlated with the soil-air TGM fluxes at all the plots (Fig. S11 and S12)."

[Figure]

**Fig. S11.** Correlations between the gradient of Hg(0) concentrations between surface soil pore (at 3

cm) and atmosphere values and soil-air TGM flux at four plots at the subtropical forest.

[Figure]

**Fig. S12.** Correlations between the gradient of Hg(0) concentrations between surface soil pores at (3 cm) and atmosphere values and soil-air TGM flux at four plots at the temperate forest.

***Comment 10:***

Line 267: were

Response: The text has been revised accordingly.

***Comment 11:***

Lines 273-279: The influence of soil humidity is discussed here only in terms of physicochemical terms. It is well known that microbiological processes in dry soils are greatly enhanced by occasional precipitation.

Response: We have added the text to the discussion about microbiological processing of Hg in line 301-305:

"Additionally, given that Hg conversion to Hg(0) in soil profiles occurs mainly via biotic processes, maximum aerobic microbial activity has been delineated at soil water content equivalent to 60% of a soil's water holding capacity (Breuer et al., 2002;Kiese and Butterbach-Bahl, 2002).

Appropriate soil moisture in the wetland would likely enhance the microbial reduction of Hg(II) to

Hg(0)."

**Comment 12:**

Line 296: "physicochemical properties" – what about microbiological ones?

Response: We have added the microbial community in the sentence in line 322-325:

"The forest canopy not only influences the soil Hg concentration by mediating atmospheric Hg deposition (Zhou et al., 2018;Zhou et al., 2017), but also alters soil physio-chemical properties (e.g.

SOM, pH, porosity) (Mo et al., 2011) and microbial communities (Nagati et al., 2020), which affect soil-air exchange."

**Comment 13:**

Line 368: Photo-reduction of Hg2+ may be a major driver in waters but hardly in soils which are impenetrable to solar radiation. More plausible is the explanation by higher soil temperature and the related higher microbiological activity.

Response: We agree with the reviewer. We have changed the text to "soil water Hg(II) to volatile

Hg(0)" in line 400.

Additionally, we have also highlighted the role of biotic processes in soil Hg(0) emissions in line

424-428:

"The Hg(0) in soil pore gas mainly results from biotic production. For example, soil sterilization can decrease Hg converted to Hg(0) by ~50% ; additionally, 1% of the soil Hg is converted to Hg(0)

via abiotic processes, compared to 6.8% by biotic processes at 283 K, and the fraction of Hg reduction by biotic processes increases with temperature increases (Pannu et al., 2014)."

**Comment 14:**

Paragraph, lines 417-433, Figure 5: Are these correlations made with data from all seasons? I

would expect different compensation points for different seasons.

Response: Yes, the correlations were made with data from all seasons.

**Comment 15:**

Figures 3 and 4: What does the x axis mean?

Response: the x axis means date, and we have added "date" in all the subfigures.

***Comment 16:***

Figures 5 and 6: Are these plots seasonally resolved? If not please state that data from all seasons were used.

Response: The data include all four seasons in each plot. We have added the note to this effects in the captions.

***Comment 17:***

SI, "Environmental measurements": The measurements of soil temperature (depth) is not mentioned here and neither in "Experimental". This parameter is the crucial one for physicochemical and microbiological processes in the soil. According to Figure S6 it seems to have been measured. I would prefer to discuss all relationships in relation to soil temperature instead of solar radiation. Solar radiation is essentially only a sort of proxy parameter for soil temperature. It is also not applicable for the night.

Response: Yes, we have measured the soil temperature by TDR in line 61-63:

"Soil percent moisture and soil temperature at 0-5 cm was monitored with Time Domain Reflectometry (TDR) Hydra Probe II (SDI−12/RS485) and a Stevens water cable tester (USA)."

Following the reviewer's suggestion, we have added some of discussion related to effects of soil temperature on Hg(0) emissions. Additionally, we have also added structural equation modeling (SEM) (Fig. 5) to evaluate the dominate factors controlling the soil-air Hg exchange flux. The results of this analysis shows that temperature is the main factor driving Hg evasion from forest soils.

***Comment 18:***

SI, description of MDL: Any information about the Hg content of litterfall and soil?

Response: We have litterfall and soil data in Chinese pine, larch, and deciduous broadleaved forest. We have added text to this effect on line 53-56 in the SI:

"From previous studies, the mean litterfall Hg concentrations were 15.8, 19.6, and 12.1 ng g$^{-1}$ in Chinese pine forest, larch forest, and mixed broad-leaved forest plots and the mean soil Hg concentrations (0-5 cm) were 72±12, 141±15, and 74±9 ng g$^{-1}$ in Chinese pine forest, larch forest, and mixed broad-leaved forest, respectively (Zhou et al., 2017)."

[revised manuscript text omitted]

and open field (e) at the subtropical forest.

**Fig. S10.** Correlations between soil moisture and air-surface Hg fluxes measured during daytime and night at the Chinese pine forest (a), larch forest (b), wetland (c), mixed broad-leaved forest (d) and open field (e) at the temperate forest.

**Fig. S11.** Correlations between the gradient of Hg(0) concentrations between surface soil pore (at 3

cm) and atmospheric values and soil-air Hg(0) flux at four plots at the subtropical forest.

**Fig. S12.** Correlations between the gradient of Hg(0) concentrations between surface soil pore (at 3

cm) and atmospheric values and soil-air Hg(0) flux at the four plots at the temperate forest.

[Figure]

**Fig. S1.** Schematic diagram of the dynamic flux chamber used in this study.

[Figure]

**Fig. S2.** Correlations between the averaged solar radiation (8:00-17:00) and air-surface Hg flux measured during daytime in Masson pine forests (a) and (b), wetland (c), evergreen broad-leaved forest (d) and open field (e) in the subtropical forest.

[Figure]

**Fig. S3.** Correlation between the averaged solar radiation (8:00-17:00) and air-surface Hg flux measured during daytime in Chinese pine forest (a), larch forest (b), wetland (c), mixed broad-leaved forest (d) and open field (e) in the temperate forest.

[Figure]

**Fig. S4.** Effects of rainfall events on annual soil-air TGM fluxes at Masson pine forests (Plot A) and
(Plot B), wetland (Plot C), evergreen broad-leaved forest (Plot D) and open field (Plot E) at the
subtropical forest (A), and at Chinese pine forest (Plot A), larch forest (Plot B), wetland (Plot C),
mixed broad-leaved forest (Plot D) and open field (Plot E) at the temperate forest (B).

[Figure]

**Fig. S5.** Correlation between the soil Hg concentrations ($S_c \pm SD$) and soil-air Hg flux ($F \pm SD$)

under the forest canopy at the subtropical forest. Standard deviations of soil Hg concentrations were obtained from Hg concentrations over the four seasons (n=12). Because fluxes are often controlled by solar radiation for bare soils, the correlation analysis above does not include data from the open field (plot E).

[Figure]

**Fig. S6.** Soil-air TGM fluxes during the daytime and nighttime at Masson pine forests (Plot A) and (Plot B), wetland (Plot C), evergreen broad-leaved forest (Plot D) and open field (Plot E) at the subtropical forest (a), and at Chinese pine forest (Plot A), larch forest (Plot B), wetland (Plot C), mixed broad-leaved forest (Plot D) and open field (Plot E) at the temperate forest (b).

[Figure]

**Fig. S7.** Correlations between soil temperature and air-surface Hg fluxes measured during daytime and night at the Masson pine forests (a) and (b), wetland (c), evergreen broad-leaved forest (d) and open field (e) in the subtropical forest.

[Figure]

s

**Fig. S8.** Correlations between soil temperature and air-surface Hg fluxes measured during daytime
and night at the Chinese pine forest (a), larch forest (b), wetland (c), mixed broad-leaved forest (d)
and open field (e) at the temperate forest.

[Figure]

**Fig. S9.** Correlations between soil moisture and air-surface Hg fluxes measured during daytime and
night at the Chinese pine forest (a), larch forest (b), wetland (c), mixed broad-leaved forest (d) and
open field (e) at the subtropical forest.

[Figure]

**Fig. S10.** Correlations between soil moisture and air-surface Hg fluxes measured during daytime
and night at the Chinese pine forest (a), larch forest (b), wetland (c), mixed broad-leaved forest (d)
and open field (e) at the temperate forest.

[Figure]

**Fig. S11.** Correlations between the gradient of Hg(0) concentrations between surface soil pore (at 3

cm) and atmospheric values and soil-air Hg(0) flux at four plots at the subtropical forest.

[Figure]

**Fig. S12.** Correlations between the gradient of Hg(0) concentrations between surface soil pore (at 3 cm) and atmospheric values and soil-air Hg(0) flux at the four plots at the temperate forest.